# SO$_2$ Layer Height retrieval from Sentinel-5 Precursor/TROPOMI using FP_ILM

Pascal Hedelt[1], Dmitry S. Efremenko[1], Diego G. Loyola[1], Robert Spurr[2], and Lieven Clarisse[3]

[1]Remote Sensing Technology institute (IMF), German Aerospace Center (DLR), Oberpfaffenhofen, Germany
[2]RT Solutions Inc., Cambridge, MA, USA
[3]Université libre de Bruxelles (ULB), Service de Chimie Quantique et Photophysique, Atmospheric Spectroscopy, Brussels, Belgium

**Correspondence:** P. Hedelt (pascal.hedelt@dlr.de)

**Abstract.** Accurate determination of the location, height and loading of SO$_2$ plumes emitted by volcanic eruptions is essential for aviation safety. The SO$_2$ layer height is furthermore one of the most critical parameters that determine the impact on the climate. Retrievals of SO$_2$ plume height have been carried out using satellite UV backscatter measurements, but until now, such algorithms are very time-consuming. We have developed an extremely fast yet accurate SO$_2$ layer height retrieval using the Full-Physics Inverse Learning Machine (FP_ILM) algorithm. This is the first time the algorithm has been applied to measurements from the TROPOMI instrument on board the Sentinel-5 Precursor platform. In this paper, we demonstrate the ability of the FP_ILM algorithm to retrieve SO$_2$ plume layer heights in near-real-time applications with an accuracy of better than 2 km for SO$_2$ total columns larger than 20 DU. We present SO$_2$ layer height results for the volcanic eruptions of Sinabung in February 2018, Sierra Negra in June 2018 and Raikoke in June 2019, observed by TROPOMI.

*Copyright statement.* TEXT

## 1   Introduction

Global satellite observations allow for the timely detection and monitoring of SO$_2$ emitted from volcanic eruptions, even in remote regions, where no ground-based instruments are installed (see e.g. Fioletov et al. 2013). Satellite measurements of UV earthshine spectra in the wavelength range between 305 and 335 nm provide the highest sensitivity to SO$_2$ in the Earth's

atmosphere. Volcanic eruptions can inject large amounts of sulphur dioxide (SO$_2$) into the atmosphere, where it is either subject to dry and wet deposition, as well as oxidization within a few days in the troposphere (see e.g. Lee et al. 2011 or Myles et al. 2011), or oxidization over a period of several weeks to sulfate aerosols in the stratosphere (see e.g. Robock, 2000; Forster et al., 2007 and von Glasow et al. 2009). Sulfate aerosols can affect the Earth's radiative forcing and have an impact on clouds (see e.g. McCormick et al., 1995; Robock, 2000 and Malavelle et al. 2017).

Based on UV earthshine measurements, SO$_2$ vertical column densities (VCDs) can be retrieved easily using for example the Differential Optical Absorption Spectroscopy (DOAS) technique, see e.g. Rix et al. (2012), a Principal Component Analysis

(PCA), see e.g. Li et al. (2017), or the Krueger-Kerr algorithm, which was applied to retrieve $SO_2$ from NASA TOMS, see e.g. Krueger et al. (1995). These methods are fast enough for near-real time (NRT) retrievals. Nevertheless, all algorithms retrieve only the slant column - in order to calculate the VCD a conversion factor (called the airmass factor, AMF) has to be applied, which includes explicit or implicit assumptions about the vertical distribution of $SO_2$ in order to determine the effective light path. Note that AMFs are calculated by means of multiple-scattering radiative transfer models assuming known vertical distributions of $SO_2$ and $O_3$, as well as viewing, surface and cloud properties.

Unfortunately, the vertical distribution (in terms of the plume layer height) of $SO_2$ is usually unknown at the time of the measurement since it is not easy to extract from the spectral signature (see Yang et al. 2009 and Nowlan et al. 2011): The $SO_2$ loading (VCD) has a direct effect on the optical depth, whereas the layer height (LH) has an indirect effect on the optical depth since it influences the number of photons passing through the $SO_2$ layer, the UV wavelengths interacting with the $SO_2$ layer, as well as the layer optical depth due to the temperature dependency of the $SO_2$ absorption crosssections (Yang et al., 2009).

Even if there are ground-based or aircraft measurements of the $SO_2$ layer-height (LH), the data are generally difficult to use for validation, since e.g. for strong eruptions (Volcanic Explosivity Index VEI$\geq$3), volcanic plumes are typically transported over long distances and the number of collocations is small. Thus, for volcanic $SO_2$ measurements, the vertical distribution of $SO_2$ is a key parameter limiting the product accuracy. The usual approach for operational $SO_2$ retrievals is to assume several different a-priori $SO_2$ vertical distributions and provide VCDs for each (see e.g. Theys et al. 2017) along with an averaging kernel (AK), in order that the user can calculate the VCD for an arbitrary $SO_2$ vertical distribution. However, in the upper troposphere and above the vertical $SO_2$ distribution has little impact on the VCD.

To date, $SO_2$ layer-height retrievals have used computationally demanding direct fitting inversion methods, which are not suitable for NRT applications. For the retrievals based on satellite UV measurements, Yang et al. (2009) and Yang et al. (2010) developed an Extended Iterative Spectral Fitting (EISF) algorithm for the Ozone Monitoring Instrument (OMI) aboard the NASA Aura satellite, and Nowlan et al. (2011) introduced an optimal estimation (OE) scheme for the GOME-2 instrument (Global Ozone Monitoring Experiment-2) aboard the EUMETSAT/ESA MetOp satellite fleet. For strong volcanic eruptions (VEI$\geq$3), the accuracy of the retrieved $SO_2$ LH using this approach is in the range 0.5–1 km, whereas for small $SO_2$ absorption it is around 2 km (see Nowlan et al. 2011).

Satellite infrared sounders also offer the opportunity to measure both $SO_2$ VCDs and LH (see Clarisse et al., 2008; Carboni et al., 2012; Clarisse et al., 2014 and Carboni et al. 2016) using optimal estimation algorithms for the Infrared Atmospheric Sounding Interferometer (IASI) aboard the EUMETSAT/ESA MetOp satellite fleet. While infrared sounders have a inferior sensitivity to the lower tropospheric $SO_2$ than that for UV instruments, the layer height retrievals tend to have a better accuracy, and perform well even for low column amounts (up to the single-DU level, see e.g. Clarisse et al. 2014 or Carboni et al. 2016.), however, the horizontal resolution (12 km), is rather coarse.

In contrast, the TROPOMI instrument on board the Sentinel-5 Precursor satellite (S5P) launched on 13 October 2017 has a much higher spatial resolution of $7 \times 3.5\,km^2$, and will operate at an even smaller resolution of $5.5 \times 3.5\,km^2$ beginning mid-August 2019. This allows us to observe and study $SO_2$ plumes at an unprecendented level of detail. Data turnover from

TROPOMI is very large, and this consideration will require the development of new retrieval schemes for the fast and accurate retrieval of $SO_2$ layer heights in an operational environment.

To this end, we have developed an algorithm called 'Full-Physics Inverse Learning Machine' (hereafter referred to as FP_ILM) for the retrieval of the $SO_2$ LH based on satellite UV earthshine spectra. The FP_ILM algorithm has been used for the retrieval of ozone profile shapes (Xu et al., 2017), the retrieval of surface properties accounting for bidirectional reflectance distribution function (BRDF) effects (Loyola et al., 2019), and the retrieval of $SO_2$ LH from GOME-2 (Efremenko et al., 2017). The algorithm creates a mapping between the spectral radiance and $SO_2$ LH using machine learning methods. The time-consuming training phase of the algorithm using radiative transfer model calculations is performed off-line, and only the inversion operator has to be applied to satellite measurements - this makes the algorithm extremely fast and it can thus be used in near-real time processing environments. In this second paper on the FP_ILM $SO_2$ LH, we describe some improvements to the original algorithm from Efremenko et al. (2017), and we apply it to a number of volcanic eruptions observed by TROPOMI during the operational phase of the mission (started April 2018).

The paper is organized as follows: In Section 2 we describe the improved FP_ILM $SO_2$ LH algorithm. The sensitivity of retrieved $SO_2$ LHs to a number of different parameters is discussed in Section 3. In Section 4, the FP_ILM is applied to S5P data to retrieve $SO_2$ LHs for selected volcanic eruptions. Section 5 describes how the algorihm could be integrated in the operational TROPOMI $SO_2$ VCD retrieval algorithm. We summarize the paper in Section 6.

## 2 FP_ILM algorithm

Conceptually, the FP_ILM consists of a training phase, in which the inversion operator is obtained using synthetic data generated with an appropriate radiative transfer (RT) model, and an operational phase, in which the inversion operator is applied to real satellite measurements. The main advantage of the FP_ILM over classical direct fitting approaches is that the time-consuming training phase involving complex RT modeling is performed offline; the inverse operator itself is robust and computationally simple and therefore extremely fast. In our previous paper (see Efremenko et al. 2017), we first introduced the FP_ILM algorithm and applied it to GOME-2 observations of a number of volcanic eruptions. We used a combination of principal components analysis (PCA) and principal component regression (PCR) methods to train the inversion operator in order to retrieve the $SO_2$ LH. For the current paper we have improved the FP_ILM algorithm with the use of a neural network (NN) approach, as outlined below.

During the training phase, the LInearized Discrete Ordinate Radiative Transfer model (LIDORT) with inelastic rotational Raman scattering (RRS) implementation (Spurr et al., 2008) is deployed to compute simulated reflectance spectra in the wavelength range $310 - 335$ nm. These spectra depend upon the following $n = 8$ input parameters: the $SO_2$ VCD and LH, the surface albedo, the surface height, the $O_3$ VCD, the solar zenith angle (SZA), the viewing zenith angle (VZA) and the relative azimuth angle (RAA). Table 1 provides a summary of the final parameter space after optimization (see below) used for the training of the final retrieval operator. Note that $O_3$ has to be included due to the strong spectral interference between $SO_2$ and $O_3$ in the spectral range considered. $O_3$ profiles are classified according to the total column amount of ozone, and the

month and latitude zones as specified in the TOMS Version 8 $O_3$ profile climatology (Bhartia, 2003). The $SO_2$ plume profile is taken to have a Gaussian shape, characterized by the total $SO_2$ VCD loading and centered at a peak-concentration layer height $z_p$, along with a half-width fixed to 2.5 km. In the following, the retrieval of $SO_2$ layer height refers to the retrieval of the peak-concentration height $z_p$.

Simulations were done on a pressure/temperature/height grid from the US standard atmosphere, with a finer-grid vertical height resolution of 0.25 km below 15 km in order to resolve properly the Gaussian $SO_2$ plume shape. In total, some 131,072 simulated reflectance spectra have been calculated on a selective parameter grid established by means of a smart sampling technique developed by Loyola et al. (2016). Further details on the smart sampling technique applied to the SO2 LH retrieval can be found in Efremenko et al. (2017).

The use of the LIDORT-RRS RT model is necessary, as it enables us to account for the effect of Raman scattering in the atmosphere: Solar irradiances exhibit strong Fraunhofer structures in this part of the UV spectral range, and earthshine spectra are characterized by the "filling-in" of Fraunhofer-solar and telluric-absorber features due to "inelastic" (wavelength-redistributed) rotational-Raman scattering by air molecules. For further details, see Efremenko et al. (2017) and Spurr et al. (2008).

The simulated high-resolution reflectance spectra are convolved with the TROPOMI Instrument Spectral Response Function (ISRF) v3.0.0 (released 2018-04-01)[1]. Note that the TROPOMI instrument comprises 450 rows, which are in principle single detectors with their own ISRFs. The signal-to-noise ratio (SNR) is about 1000 in our UV wavelength range. Thus, to account for instrumental noise in the training phase, uncorrelated Gaussian noise with a fixed SNR of 1000 is added to the simulated spectral data.

To extract the information about the layer height and to reduce the dimensionality of the spectral dataset, a PCA is applied to the simulated spectra. By thus characterizing the set of simulated measurements with fewer parameters, a simpler, more stable and computationally efficient inversion scheme can be realized. It was found that using 10 principal components (PCs) is sufficient to retrieve information about the $SO_2$ LH. These 10 PCs account for 99.994% of the spectral variance. The inclusion of additional PCs beyond 10 did not result in any improvements to LH retrievals, since higher-order PCs are increasingly
affected by noise. Figure 1 shows the ratio between the variance of the PCA-derived data set and the total variance of the complete spectra data set (the "explained" variance ratio) as a function of number of PCs included in the PCA.

The 10 PCs, together with the information about the $O_3$ VCD, the SZA/VZA/RAA angles, the surface pressure and albedo of each training data-point, are then used as input to train a feed-forward artificial neural network (NN) including regression (in our case a MultiLayer Perceptron Regression - MLPR), with the corresponding $SO_2$ LH as the output layer. Note here that
the $SO_2$ VCD is not part of the training, since it depends directly on the $SO_2$ layer height.

In general, NNs can be used for establishing a non-linear mapping between a dataset of numeric inputs and a set of numeric outputs. A NN consists of interconnected neurons (or nodes) that implement a simple, non-linear function (a sigmoid function in our case) of the inputs. Neural networking is a powerful tool for determining non-linear dependencies between datasets in

---

[1]Available here:

http://www.tropomi.eu/data-products/isrf-dataset/

remote sensing as shown by Loyola (2006). It consists of an input layer (representing the above mentioned input parameters), at least one so-called hidden layer and an output layer (representing the expected output - in our case the $SO_2$ LH). Each neuron in the hidden layer(s) transforms the values from the previous layer(s) with a weighted linear summation, followed by a non-linear activation function, which is in our case the sigmoid function. The output layer receives the values from the last hidden layer and transforms them into output values.

The training of the MLPR is an iterative process that tries to minimize the so-called loss-function (also known as the cost-function), which is a measure of how well a model predicts the expected outcome; at each time step, the partial derivatives of the loss-function with respect to the model parameters are computed in order to update the parameters. We note here that MLPR uses the "mean square error" loss-function. A regularization term is added to the loss-function that shrinks model parameters to prevent over-fitting: By building a complex neural network, it is quite easy to perfectly fit the training dataset. When this model is however evaluated on new data (here the satellite measurements), it performs very poorly. The regularization thus modifies the loss function by adding additional terms that penalize large weight vectors and preferring diffuse weight vectors.

After carrying out a PCA and MLPR parameter optimization using closed-loop retrievals to minimize differences between the retrieved and simulated layer heights, the final configuration for the neural network settled on the use of two hidden layers, with 32 nodes in the first layer and 10 nodes in the second.

In the operational phase, the first step is to use the principal component scores acquired during the training phase to transform a given TROPOMI spectral measurement data set to one with a lower dimension. Once this is done, the neural network inverse function is then applied to retrieve the $SO_2$ LH. In order to avoid the training of the NN for each of the 450 TROPOMI detector rows (with their own ISRF), we have trained the network only for every 50th detector row and interpolate the retrieved $SO_2$ LH results to the actual row where it was detected.

We note that the FP_ILM algorithm only needs to be re-trained when large changes in the ISRF or SNR occur, see Section 3.

## 3 Dependencies

In this section, we study the dependency of the layer height retrieval on different parameters. We discriminate between direct dependencies (i.e. those parameters affecting the reflectance spectra) and indirect dependencies (i.e. affecting the training data and inversion algorithm) of the retrieved layer height.

- Direct dependencies: Viewing geometry, surface properties, ISRF, noise level (SNR), instrumental stray light, $O_3$ VCD

- Indirect dependencies: Number of layers in the neural network, number of PCs, parameter ranges for training

First, we train the FP_ILM operator using 90% of the training dataset (see Table 1), and then apply the trained operator to the remaining 10% training spectra, for which we know the exact $SO_2$ LH. Figure 2 (red dots) shows the $SO_2$ LH difference as a function of solar/viewing geometry, $SO_2$ VCD, $O_3$ VCD, albedo and surface pressure, respectively. The figure shows clearly

a number of marked dependencies in the retrieved layer height, with notably high differences with respect to the real $SO_2$ LH for low and high layer heights, for low $SO_2$ VCD well as for high SZA.

Regarding the SZA dependency, a cutoff limit of 75° is usually set in operational $SO_2$ retrievals, because at high SZA the light path becomes very long and the noise level increases. Accordingly, we limit the SZA of the spectra used in the training to SZA≤75° in the following.

Clearly, for small $SO_2$ VCDs, the information content on LH in the spectral signature is very low. It follows that the inclusion of spectra with low SO2 content in the training will have have a negative effect on the entire neural network. In principle for lower $SO_2$ VCD loadings, more PCs could be included, but at some point the noise level signature will exceed that of the actual $SO_2$ absorption.

We have performed several tests in which we limit the training dataset by varying the allowed input parameter ranges. We have found an optimal parameter range that allows the retrieval of a broad range of $SO_2$ LHs even for low $SO_2$ VCDs. In the training phase, we use only spectra with $SO_2$ VCD≥20 DU, surface albedo < 0.5 and SZA≤75° to train the final inversion operator for TROPOMI. The albedo limit is set to 0.5, since large values of the albedo will induce large variations in the spectra (multiple reflections from the surface). These additional variations correlate with the variations due to perturbations in the $SO_2$ VCD and LH. Therefore, to make the algorithm more stable, we restrict the albedo range to the physically relevant cases. We have further limited the training values of $SO_2$ LH to the range 0–25 km. Although higher LHs from strong volcanic eruptions can occur, the use of a broad training data range also has an influence on the accuracy of the retrieval. To a limited extend however, the FP_ILM is also able to extrapolate to an untrained parameter range, however with significantly less accuracy.

Using the optimized training dataset (see Table 1 for a summary of the parameter space), Fig. 2 (blue triangles) shows that the error on the retrieved $SO_2$ LH is less than 2 km and the dependency on $SO_2$ VCD is reduced. Figure 3 shows the retrieved layer height as a function of $SO_2$ VCD. As mentioned above, for low $SO_2$ VCDs, high-altitude layer heights cannot be retrieved - there is always a bias towards low layer heights. Only for $SO_2$ loadings in excess of 20 DU, do we retrieve layer heights with an uncertainty of less than 2 km.

To investigate the dependency of the retrieved $SO_2$ LH as a function of SNR, we have used two different noise levels (SNR of 500 and 1500). Figure 4 clearly shows that the SNR only has a minor effect on the accuracy of the retrieved layer height, with only slightly higher accuracy for increased SNR. This is to be expected, since we have used the first 10 PC scores and thus basically removed all noise features.

Concerning the dependency on the ISRF, we note that in the operational retrieval of $SO_2$ LHs from TROPOMI data, each detector row is effectively a single instrument with its own ISRF. The accuracy of the retrieved $SO_2$ LH can vary across detector rows. Figure 5 shows the $SO_2$ LH results when applied to the Sinabung eruption (see Section 4.1). Black dots show the LH for each 50th row, whereas the red cross shows the interpolated LH for the measurement row. Clearly, the retrieved LH is slightly different in each row (within 1 km) due to the different ISRF used for training the NN. Note that we determine the layer height for a set of fixed detector rows (for which we have trained the FP_ILM separately), and then interpolate the layer height to the actual row. In this way we avoid jumps in the retrieved layer height between adjacent detector rows.

Instrumental straylight can introduce spectral features that may lead to bias in the retrieved $SO_2$ LH. However, according to Kleipool et al. (2018) the in-band straylight of TROPOMI after correction is as low as 0.5% and thus can be neglected. Furthermore no evidence for out-of-band-straylight was found (pers. comm. Quintus Kleipool).

During major volcanic eruptions (VEI$\geq$3) with very high $SO_2$ loadings, the $O_3$ VCD retrievals may be inaccurate due to $SO_2$ interference. However, this effect is negligible for weak eruptions. For strong eruptions (exceeding about 50 DU of $SO_2$) the error on the $O_3$ VCD may reach a few percent (see Lerot et al. 2014) and hence the error on the $SO_2$ LH is negligible.

## 4 Application to TROPOMI data

Reflectance spectra from TROPOMI are determined from the operational L1 solar irradiance and earthshine radiance data (solar irradiance is measured on a daily basis). To correct for Doppler shifts between earthshine and irradiance spectra, we apply the wavelength calibration information from the operational L2 $SO_2$ product to first register the solar spectrum and then we use the fitted shift and squeeze parameters from the DOAS retrieval to calibrate the earthshine spectra. From this calibrated L1 data, we then calculate reflectances in the wavelength range $310 - 335$ nm.

In the following subsections, we have applied the FP_ILM operator to three major volcanic eruptions measured by TROPOMI. We have chosen eruptions having a peak $SO_2$ VCD exceeding our 20 DU threshold criterion and with an extended $SO_2$ plume that allows us to compare the FP_ILM results with independent retrievals from other satellite data sources. In addition to L1 reflectances, we use information on $SO_2$ VCD, $O_3$ VCD, surface and viewing conditions from the operational $SO_2$ L2 product, see Pedergnana et al. (2018).

For validation of our results, we have performed comparisons with independent MetOp/IASI $SO_2$ LH data (see Clarisse et al. 2014 for details), as well as $SO_2$ profile data from the Microwave Limb Sounder (MLS) on the NASA/Aura satellite (see Pumphrey et al. 2015). The MetOp platforms have different orbits from that of S5P, with widely different overpass times, so that direct satellite comparisons with IASI data will give only a qualitative validation on the accuracy of retrieved $SO_2$ LHs from TROPOMI measurements. The afternoon Aura/MLS overpass is nearly coincident with TROPOMI. MLS can provide some information on $SO_2$ LH, albeit with limited spatial coverage and vertical resolution, since it is only sensitive above about 147 hPa altitude (i.e. above around 13 km). We also checked on the availability of an overpass of the Cloud-Aerosol Lidar and Infrared Pathfinder Satellite Observation (CALIPSO, Winker et al. 2010) satellite with the the Cloud-Aerosol Lidar with Orthogonal Polarization (CALIOP) instrument. Note that CALIPSO measures only ash and aerosol absorption profiles. In this regard, we note that $SO_2$ plumes and ash or aerosols are not necessarily collocated, as gas and ash can separate in volcanic clouds. At the time of writing, IASI and MLS data are unfortunately the only sources for independent $SO_2$ LH satellite validation.

## 4.1 Sinabung

The Sinabung stratovolcano (2,460 m summit elevation) on the island of Sumatra has been highly active until September 2010 and was quiet until a new eruptive phase began in September 2013 which lasted until March 2018, see Venzke (2018b). On 19

February 2018, Sinabung erupted violently at around 02:55h UTC with its largest explosion to date, emitting a volcanic ash plume that rose to at least 16.8 km altitude and an $SO_2$ plume of up to 50 DU of $SO_2$ that was observed by several satellite instruments, including TROPOMI (see Fig. 6, overpass time around 06:30h UTC), OMI, IASI (overpass time around 03:30h UTC and 15:00h UTC), MLS (overpass time around 07:10h UTC). There was also an overpass of CALIPSO/CALIOP at about 07:15h UTC, over the volcanic plume.

The FP_ILM algorithm retrieved $SO_2$ LHs extending up to 17 km, see Fig. 6; these show excellent agreement with the MLS $SO_2$ profile measurements inside the plume (see colored circles in Fig. 6), with peaks at 16.75 km close to the volcano, and 14.31 km in the northern part of the plume, see Fig. 8. Furthermore, our results are in close agreement with CALIPSO measurements, clearly showing an attenuation by ash or aerosols at altitudes around 15-18 km (color-coded in orange). Fresh volcanic plumes are typically rich in water vapour (especially for tropical eruptions), and thus the volcanic clouds also contain high concentrations of water droplets. Therefore, the classification in the CALIPSO vertical feature mask sometimes fails to pick up the volcanic ash or sulfate aerosol because of competing clouds. Nevertheless the altitudes of the identified features are likely those of the volcanic plumes themselves, given the collocations in time and space. We note that the brightness temperature difference plots (not shown) can help in the identification of an ash layer, since the difference in the brightness temperature between the CALIPSO channels at $10.6\mu$m and $12.05\mu$m becomes negative for ash, whereas for normal clouds it is positive.

In the figure we have overplotted our FP_ILM $SO_2$ LH results in red dots. $SO_2$ LH retrieved for IASI/MetOp-A and -B measurements (not shown) indicated LH values at about 13 km for the 03:30h UTC overpass and LH up to 18 km around the overpass time of 15:00h UTC; these results agree well with the TROPOMI results.

Also the Pusat Vulkanologi dan Mitigasi Bencana Geologi[2] (PVMBG, also known as CVGHM) reported at 08:53h UTC *a dark gray plume with a high volume of ash that rose at least to 16.8 km*. Furthermore the Darwin Volcanic Ash Advisory Center[3] (VAAC) reported that LH values for these Sinabung ash plumes identified in satellite images, recorded by webcams, and reported by PVMBG continued to rise throughout the day to 13.7 km.

## 4.2 Sierra Negra

On 26 June 2018 a strong eruption at the Sierra Negra shield volcano (1,124 m summit elevation) located on Isabela Island (Galapagos) occurred. According to Venzke (2018a), this volcano has erupted several times since 1948, with the last eruption reported in 2005. After an increase in seismicity, Sierra Negra erupted at 20:09h UTC, producing a dense ash and $SO_2$ plume. The eruption was divided into an initial very energetic phase (VEI = 3) characterized by the opening of five fissures that lasted one day, and a long phase with lava flows from 27 July to 23 August 2018 with decreased gas emissions. TROPOMI was able to measure a very strong $SO_2$ plume (with loading in excess of 500 DU) only a few minutes (overpass at 20:12h UTC) after the start of the first eruption, as well as an extended $SO_2$ plume for several days after the eruption.

---

[2]see http://vsi.esdm.go.id/
[3]see http://www.bom.gov.au/aviation/volcanic-ash/

Figure 9 shows the SO$_2$ plume from 26 until 28 June at overpass times around 20:12h UTC. The corresponding FP_ILM SO$_2$ LH is shown in Fig. 10. On 26th June, shortly after the eruption began, the retrieved SO$_2$ LH was around 4–6 km. On the following days however, a much higher SO$_2$ LH of around 14 km was retrieved by FP_ILM, with some parts of the plume reaching about 18 km altitude. We note here that these high SO$_2$ LH values are not visible in the figure since they were retrieved for SO$_2$ VCD of about 10 DU and hence with low accuracy - in the figure only SO$_2$ LH results for SO$_2$ VCD$\geq$20 DU are shown.

We found only one closely related MLS SO$_2$ profile measurement intersecting the SO$_2$ plume on 27th of June, at a measurement time of 20:16h UTC, i.e. shortly after the TROPOMI measurement (see colored circle in Fig. 10 bottom, center). The MLS data in Figure 11 shows an SO$_2$ layer in the altitude range 9–12 km, which is in excellent agreement with the FP_ILM SO$_2$ LH results of around 11-13 km at the same coordinates (see zoom-in).

Figure 12 shows the SO$_2$ LH retrieved by IASI from 27 until 29th of June. Note that the IASI overpass is around 02:00h UTC and hence about 6 hours after the TROPOMI overpass. Nevertheless the LHs retrieved close to the volcano are in very good agreement with the FP_ILM results, being also at about 4-6 km. Similarly for the extended plume, for which only a few pixels are shown in Fig. 10, the agreement is very good, with LHs of about 13 km.

### 4.3 Raikoke

On 22 June 2019 at 04:00h local time, the Raikoke stratovolcano located on the Kuril Islands (Russia, 551 m summit elevation) erupted explosively (VEI $\geq$ 4) after being dormant since 1924. There were several strong distinct explosions, producing a dense ash and SO$_2$ plume that rose until 13 km altitude the first days and was entrained into the stratosphere (see Sennert 2019). This was the strongest volcanic eruption since the Merapi eruption in 2011, producing a colossal SO$_2$ plume with an SO$_2$ loading of more than 900 DU on 22 June 2019, that was dispersed by strong winds over Russia and North America, and was detectable even two weeks after the volcanic eruption. The Raikoke eruption was still ongoing during the writing of this paper. Further results for this eruption are expected to be subject of a future publication.

Figure 13 shows SO$_2$ VCD measured by TROPOMI for the first three days after the eruption. Note that the SO$_2$ plume is close to the date line with several overlapping S5P orbits at different overpass times. Therefore we have chosen to plot only one single orbit per day in the images. The figure shows the plume at overpass times of around 00:00h and 02:00h UTC.

Figure 14 shows the corresponding SO$_2$ LH retrieved by the FP_ILM for SO$_2$ VCD$\geq$20 DU. Clearly, the plume shows several layers, with SO$_2$ LH ranging from 6–8 km up to 19 km on 23 June and from 11 km up to 20 km on 24 June 2019. This is in very good agreement with MLS data (colored circles in the figure) with overpass times around 02:20h UTC on 23 June (left plot), around 01:30h UTC on 24 June (center plot) and around 00:30 UTC on 25 June (left plot): The MLS profiles in Fig. 15 show an SO$_2$ layer at 17 km on 23 June, two distinct layers at 12 and 17 km at different positions of the plume on 24 June and a layer in the range from 12–14 km on 25 June.

In addition, CALIOP/CALIPSO was able to detect the ash plume during the first days after the eruption, with plume height values in very good agreement with the FP_ILM SO$_2$ LH results. On 22 June an ash plume at 17 km was detected (not shown here), The CALIPSO data from 23 June (overpass around 01:30h UTC) shown in Fig. 16 (left) show an ash layer around 5–8 km in the northern part of the plume (i.e. around 46°N), and a second ash layer around 13 km in the southern part, in very

good agreement with the FP_ILM SO$_2$ LH results. On 24 June an ash layer at 13 km is visible which is in agreement with the lowest SO$_2$ LHs retrieved. Note that the CALIPSO ground tracks are shown in Fig. 14 (blue line).

The latest IASI retrievals indicate that the bulk of the SO$_2$ mass is contained within 9–15 km (not shown here). We note that at the time of writing this paper, the IASI LH retrieval algorithm is undergoing improvements to handle cases with very large SO$_2$ loadings in the center of the plume. Since the Raikoke eruption was still ongoing during the writing of this paper, we refer here to an upcoming publication of the IASI results for the Raikoke eruption.

## 5  Implementation in an operational environment

In this section we describe the manner in which the algorithm could be implemented in the operational S5P/TROPOMI ground segment.

Since the same input parameters are used for retrieving operational SO$_2$ VCD, i.e. O$_3$ VCD, viewing parameters and surface conditions as well as radiance and irradiance data, the FP_ILM algorithm can be easily integrated within the operational UPAS processor used for generating the SO$_2$ products. In the operational TROPOMI environment, cloud properties and the O$_3$ VCD are retrieved before the SO$_2$ algorithm is started. Thus all required input parameters are already available when the FP_ILM algorithm is triggered in the case of a volcanic eruption. Only the wavelength calibration and calculation of the reflectance spectrum has to be performed prior to retrieving the SO$_2$ LH, since this step differs from the operational SO$_2$ VCD retrieval.

To process a satellite pixel containing volcanic SO$_2$, the FP_ILM algorithm should be triggered by the operational enhanced SO$_2$ detection flag (see Theys 2018 for a reference) and using a threshold (a priori) SO$_2$ VCD of at least 10 DU. The resulting SO$_2$ LH for this pixel should be stored in the final L2 SO$_2$ product and can be also used to calculate an optimized SO$_2$ VCD for this LH.

The SO$_2$ LH retrieval only takes about 2 ms per TROPOMI spectrum, hence even for an extended volcanic plume, the entire LH retrieval can be performed in a matter of seconds. This is important for operational retrieval environments with strong time constraints. Currently, the entire processing of a single TROPOMI pixel (i.e. cloud parameters, O$_3$ and SO$_2$ VCD retrieval) in the operational ground segment takes about 90 ms, which translates to about 24 min for an entire S5P orbit. Thus the additional time required for the retrieval of the SO$_2$ LH for selected pixels is not significant.

## 6  Conclusions

We have developed a new algorithm for the fast and accurate retrieval of SO$_2$ layer heights from UV earthshine observations of volcanic SO$_2$ eruptions by the TROPOMI sensor on board the Sentinel-5 Precursor platform. The SO$_2$ LH retrieval has two phases - (1) a computationally expensive off-line training phase in which the retrieval inverse operator is obtained using the FP_ILM (Full Physics Learning Machine) algorithm; and (2) a fast operational phase, in which the FP_ILM inverse operator is applied to measured UV reflectance spectra. The FP_ILM combines a Principal Component Analysis with a Neural Network regression using the UV reflectance, O$_3$ total column, viewing geometries and surface properties as input. Based on an

optimized training dataset created with smart sampling techniques, the principal component scores calculated for reflectance spectra in the wavelength range 310-335 nm are used along with the other parameters to train a feed-forward artificial neural network. For S5P/TROPOMI measurement data, an initial dimensionality reduction of the reflectance spectra is performed by applying the PCA-derived Principal Component scores before retrieving $SO_2$ LH with the trained Neural Network.

The FP_ILM can be used for NRT applications with strict time constraints. S5P/TROPOMI, with its high spatial and spectral resolution, provides a huge amount of data and the computationally intensive direct fitting approaches to $SO_2$ LH retrieval developed so far are not applicable. In contrast, the FP_ILM operator performs $SO_2$ LH retrieval within about 2 ms per TROPOMI spectrum. Hence even the retrieval on extended volcanic plumes can be performed in a matter of few seconds, allowing for the determination of the $SO_2$ layer height with an accuracy better than 2 km for $SO_2$ total column densities larger than 20 DU.

In this paper, we deployed an independent simulated reflectance spectra dataset to investigate the accuracy of $SO_2$ LH retrievals and their dependencies on a number of different factors and parameters both direct and indirect. In particular, it was found that retrieved $SO_2$ LH is strongly dependent on the $SO_2$ VCD for low VCDs $\leq$20 DU as well as for high SZAs$\geq$75°. For high VCDs and low SZA, according to closed-loop tests, the $SO_2$ LH can be retrieved with an accuracy of better than 2 km. We also investigated the dependencies on ISRF and SNR, both of which turned out to be relatively slight effects.

Broad-band spectral scattering and absorption due to sulfate aerosols or volcanic ash plumes will certainly affect $SO_2$ LH retrievals. Although not considered in the present work, we will address this important issue in a forthcoming paper in this series; aerosols will be accounted for explicitly in the training of the FP_ILM. Nevertheless, we should note that $SO_2$ and ash are likely to be collocated only for fresh volcanic plumes. For mature plumes, mass differences will ensure that ash and $SO_2$ plumes are not located at similar altitudes, and the corresponding plumes are thus subject to different wind-direction dispersal.

We have applied the FP_ILM to a number of volcanic eruptions (VEI$\geq$3) observed recently by S5P/TROPOMI. Our $SO_2$ LH results have been compared to $SO_2$ LHs retrieved from IR measurements from two MetOp/IASI satellites, as well as NASA AURA/MLS $SO_2$ profile and CALIOP/CALIPSO ash and aerosol measurements. Unfortunately the orbits of the MetOp satellites and S5P have widely different overpass times, allowing only for qualitative comparisons. Despite this, there is in general a very good agreement between the IASI and TROPOMI results. Very good agreement with MLS data was found, 25 which provides $SO_2$ profiles with limited spatial coverage and vertical resolution, however with overpass times close to the S5P overpass. Nevertheless, further verification work is certainly needed. For the Sinabung eruption in February 2018 and the Raikoke eruption in 2019, our results also showed excellent agreement with CALIOP/CALIPSO LIDAR data measuring the ash plume.

*Acknowledgements.* We are grateful to Nicolas Theys for helpful discussions of the $SO_2$ layer height retrieval from UV sensors. We ac-
30 knowledge ESA and KNMI for the Sentinel-5 precursor L1 data and NASA for the CALIPSO and MLS data used in the paper. We would like to thank two anonymous referees for their comments that helped us improve the manuscript. We hereby acknowledge financial support from DLR programmatic (S5P KTR 2472046) for the development of TROPOMI retrieval algorithms. L.C. is a research associate supported by the Belgian F.R.S.-FNRS.

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

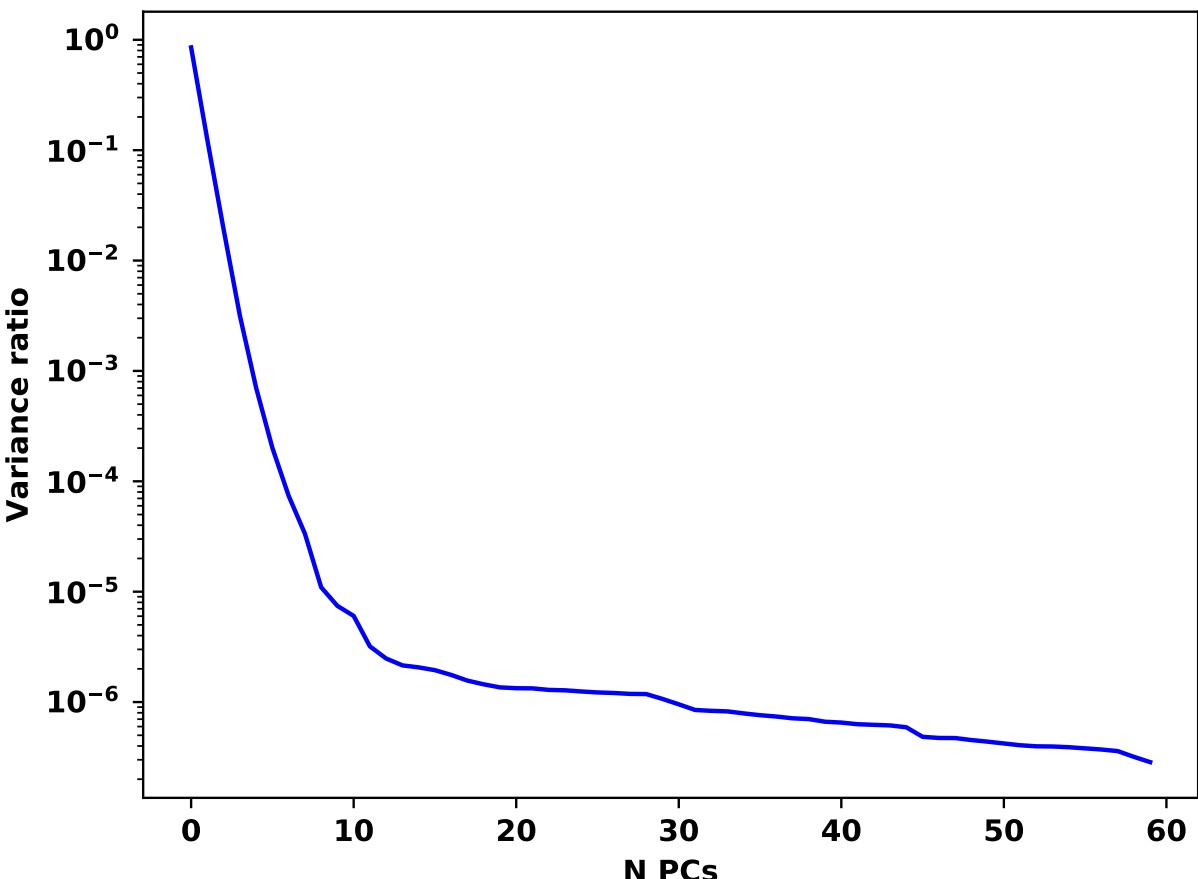

**Figure 1.** "Explained" variance ratio as a function of the number of principal components. Already the inclusion of 10 PCs account for 99.994% of the spectral variance.

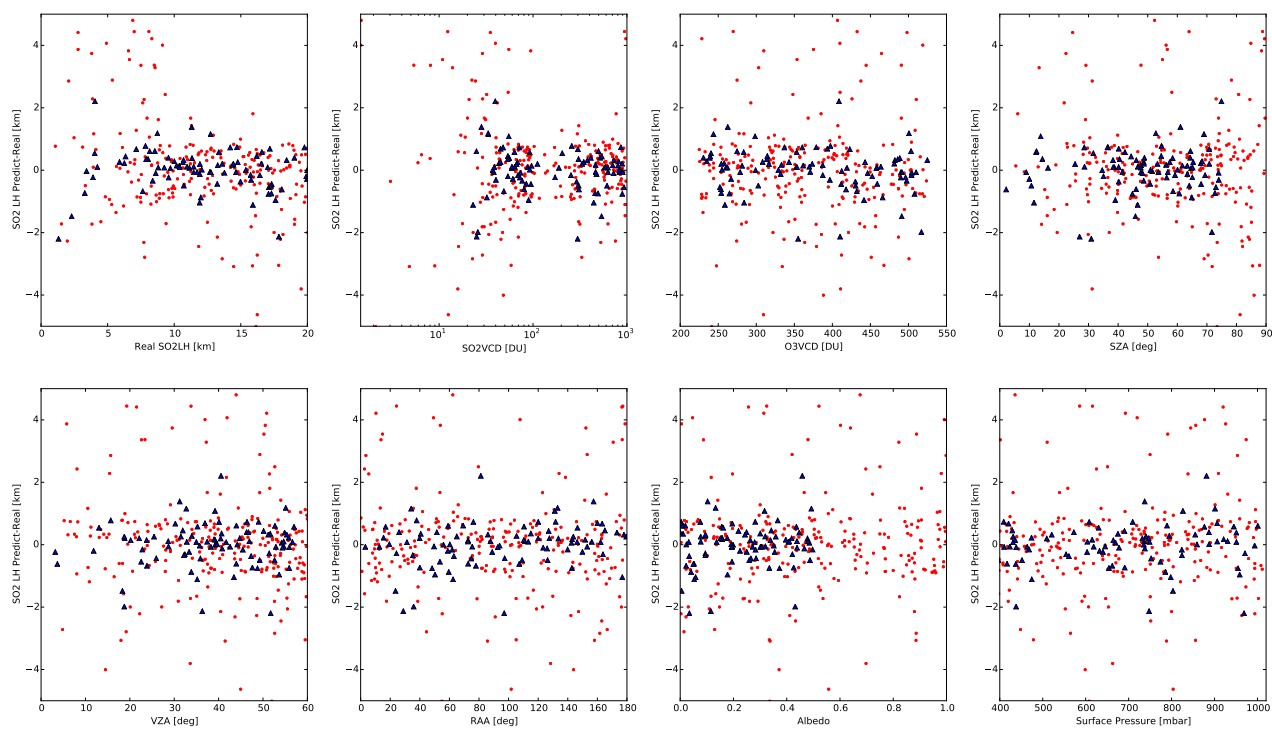

**Figure 2.** Dependency of the retrieved SO$_2$ layer height as a function of 8 parameters as indicated. Plotted are the layer height differences (in [km]) between retrieved layer heights and those simulated using the independent verification dataset (i.e. a 10% subdivision of the entire training dataset). Red dots show the dependencies using the entire training data set, whereas blue triangles show the dependencies with an optimized training dataset (i.e. restricted to SO$_2$ VCD values $\geq$20 DU, SZAs$\leq$75° and albedos $\leq$0.5.)

**Table 1.** Physical parameters varied for the generation of reflectance spectra. The optimized parameter range for training the final FP_ILM retrieval operator is shown.

| Parameter | Range |
| --- | --- |
| SZA | 0–75° |
| VZA | 0–75° |
| RAA | 0–180° |
| Surface albedo | 0–0.5 |
| Surface height | 0–8 km |
| O$_3$ VCD | 225–525 DU |
| SO$_2$ VCD | 0–1000 DU |
| SO$_2$ LH | 2.5–25 km |

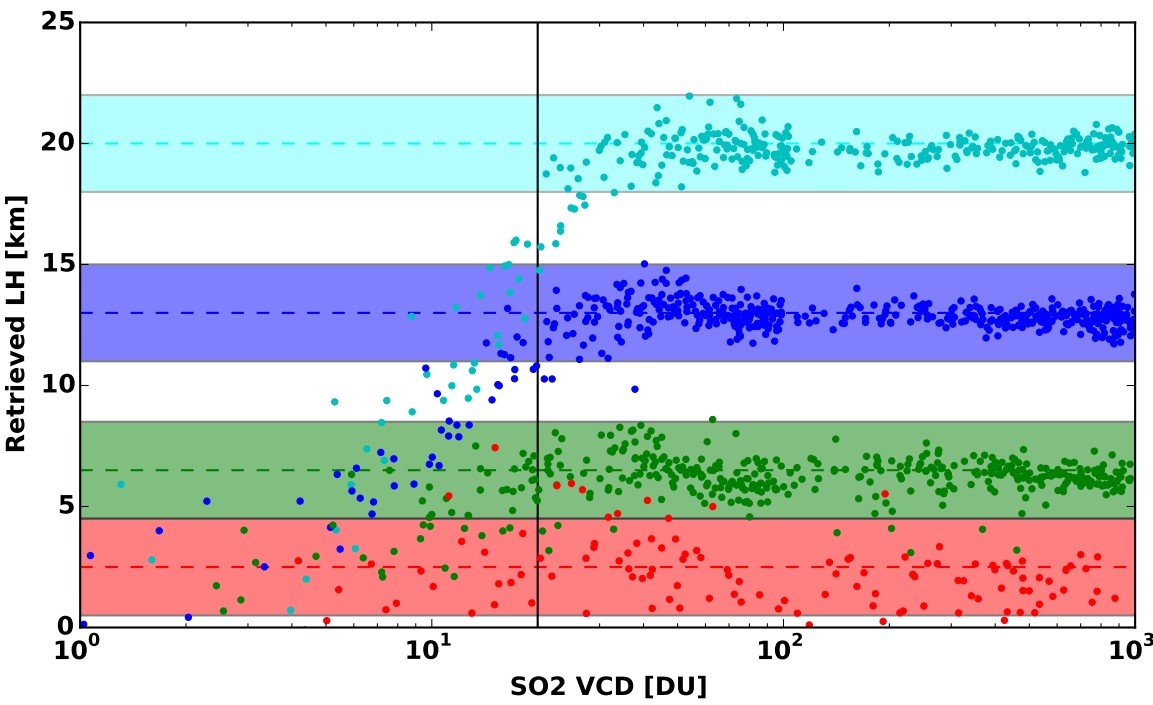

**Figure 3.** Dependency of the retrieved $SO_2$ LH as a function of $SO_2$ total vertical column, using the independent test dataset. Color coded are the different LHs for which the test spectra have been generated. Horizontal color bars indicate a 2 km uncertainty on the retrieved $SO_2$ LH.

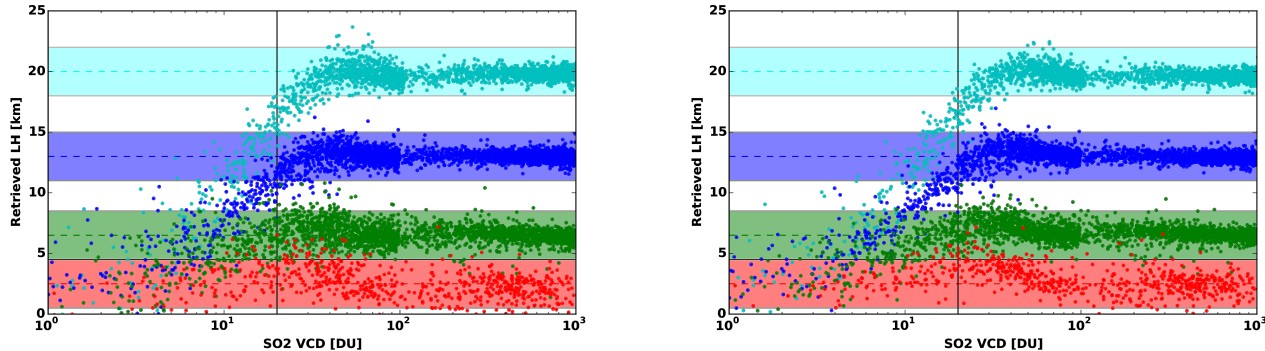

**Figure 4.** Dependency of the retrieved $SO_2$ LH on the SNR. Same as Fig. 3 with different SNR values. Left: SNR=500, Right: 1500.

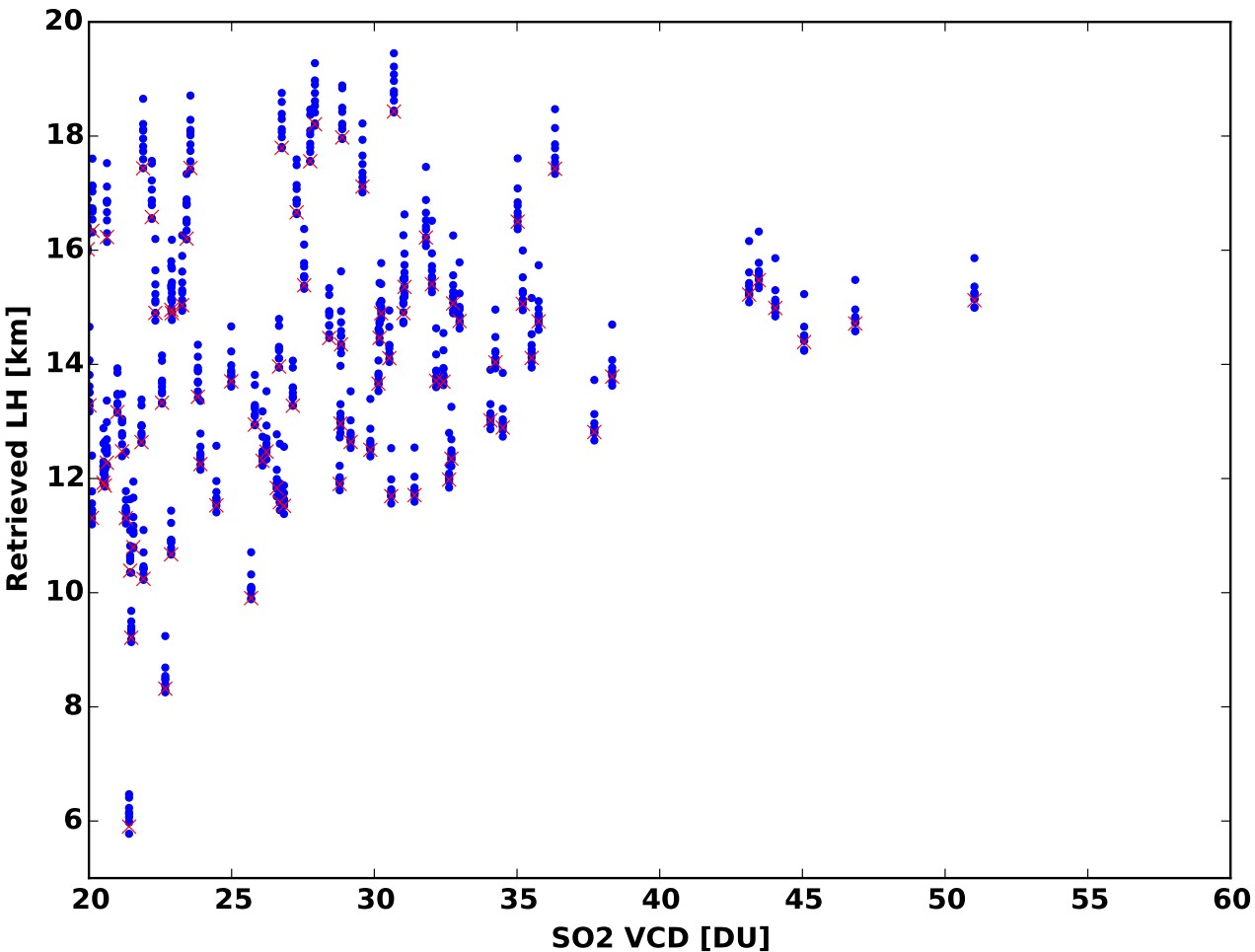

**Figure 5.** Dependency of the retrieved SO$_2$ LH on the TROPOMI instrument row. The SO$_2$ LH as a function of SO$_2$ VCD is shown for the Sinabung eruption. Blue dots show the SO$_2$ LH result for every 50th detector row, whereas red crosses show the LH interpolated to the measurement row.

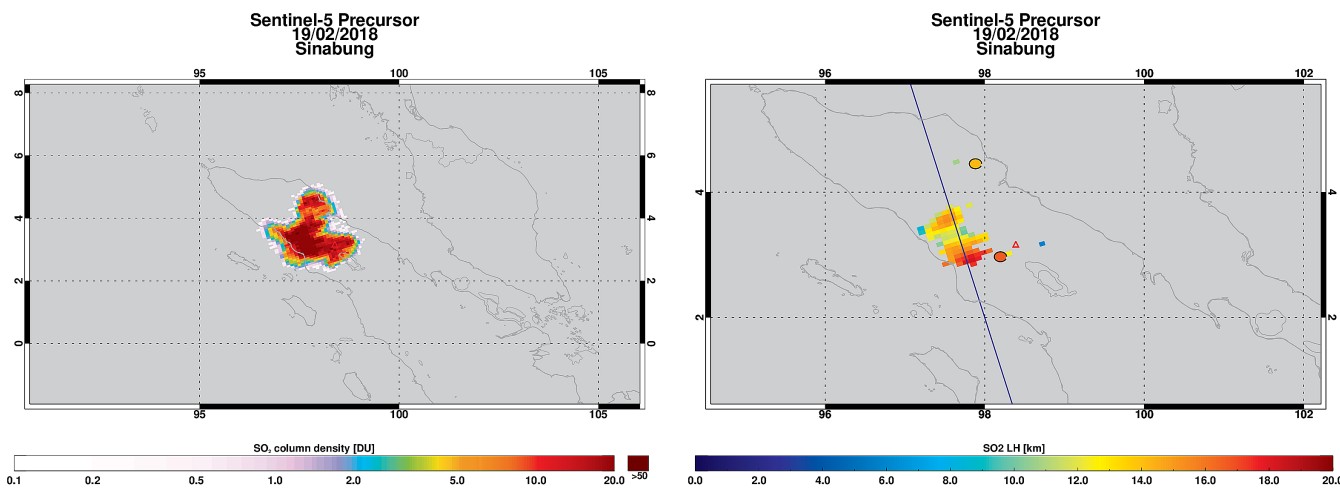

**Figure 6.** SO$_2$ VCD (left) and SO$_2$ LH (right) for the TROPOMI measurements of the Sinabung volcano on 19 February 2018, at an overpass time around 06:30h UTC. SO$_2$ LH results are only shown for VCD$\geq$20 DU. In the right plot also MLS/AURA results (the two colored circles) are shown, as well as the CALIPSO groundtrack (cf. Fig. 7)

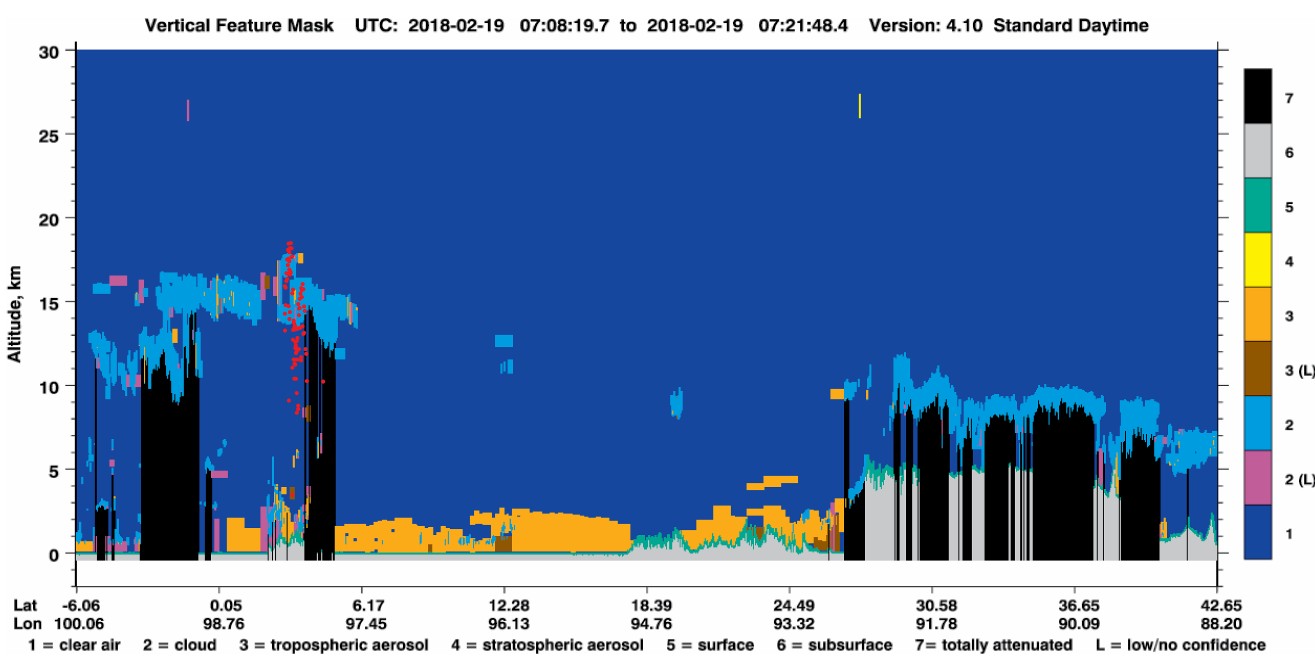

**Figure 7.** CALIOP/CALIPSO vertical feature mask for the measurement of the Sinabung volcanic eruption on 19 February 2018 around 07:15h UTC. Red dots show the retrieved LH for pixels with $SO_2$ VCD$\geq$20 DU. The ground track of the CALIPSO measurements is shown in Fig. 6. The volcano is located at 3.17N, 98.39E. The plume is clearly visible in the left part of the image. Note that the classification scheme sometimes cannot pick up the volcanic ash or sulfate aerosol because of competing clouds. CALIPSO image credit: NASA

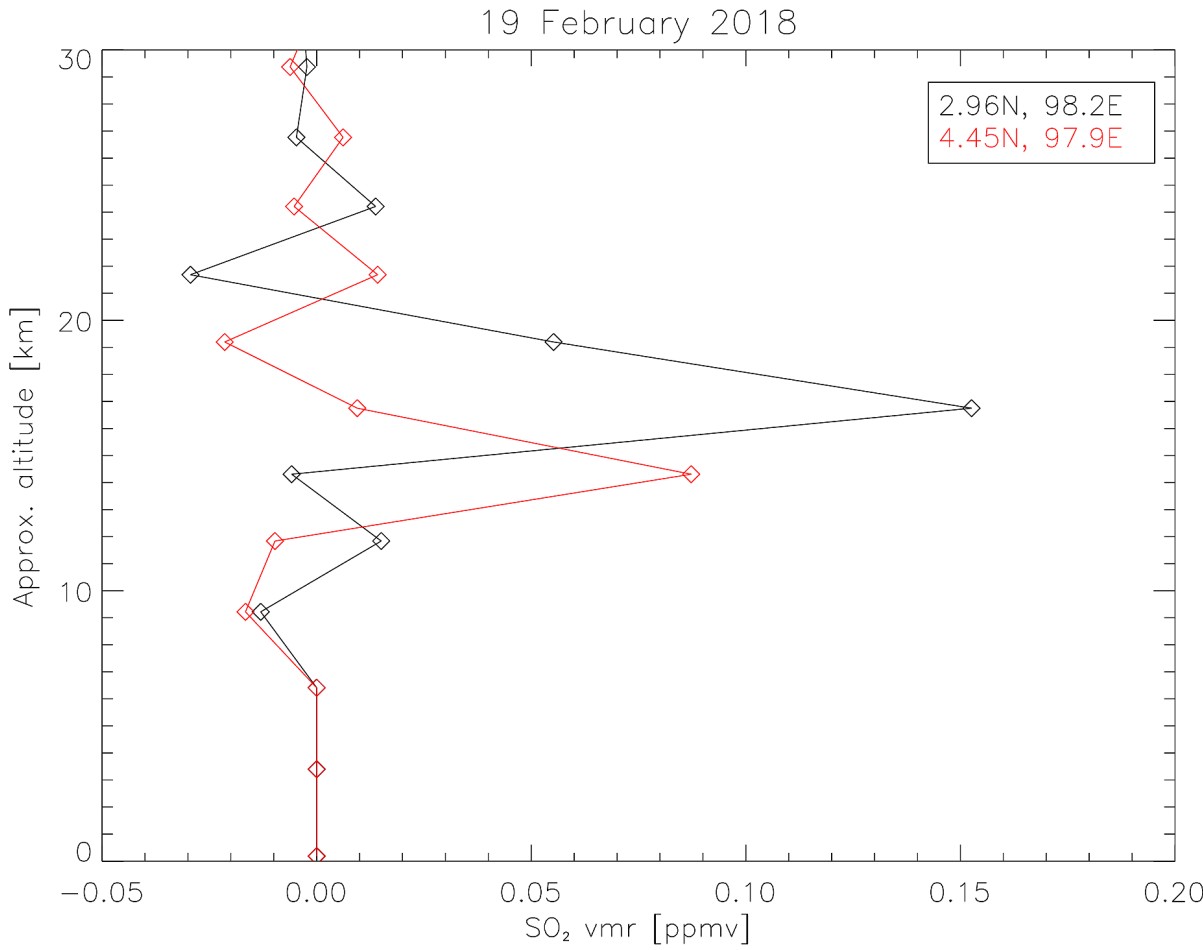

**Figure 8.** MLS SO$_2$ profile for two measurements intersecting the volcanic plume of the Sinabung volcanic eruption on 19 February 2018, clearly showing the presence of an SO$_2$ layer at an altitude around 16 km close to the volcano (black) and 13 km in the extended plume farther north (red). The position of the measurements is shown in Fig. 6.

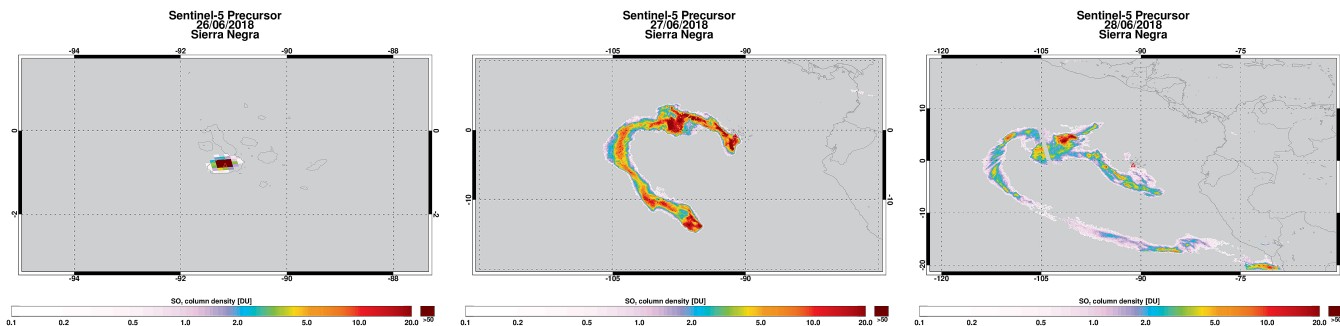

**Figure 9.** SO$_2$ VCD for the TROPOMI measurements of the Sierra Negra volcano from 26 (left) until 28 June 2018 (right), with overpass times around 19:50h UTC.

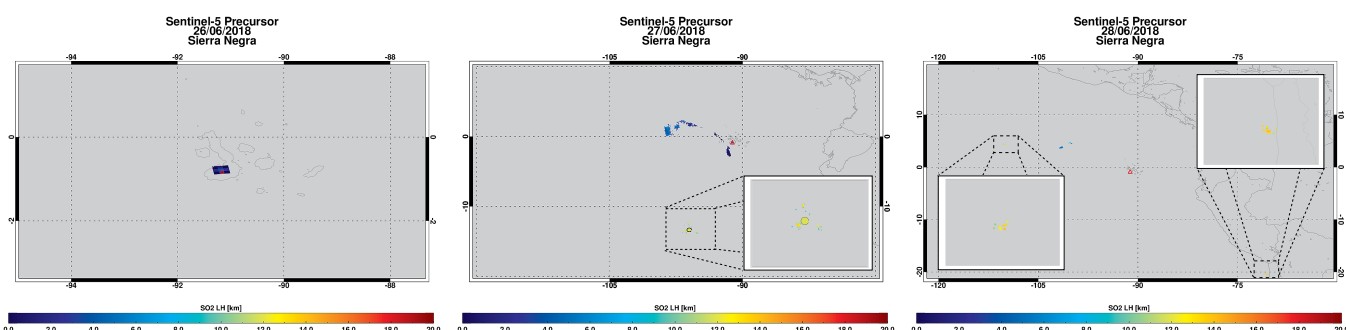

**Figure 10.** FP_ILM SO$_2$ LH for the TROPOMI measurements shown in Fig. 9. Note that only pixels with SO$_2$ VCD$\geq$20 DU are shown. The MLS SO$_2$ measurement of the volcanic plume on 27 June is indicated with a colored circle.

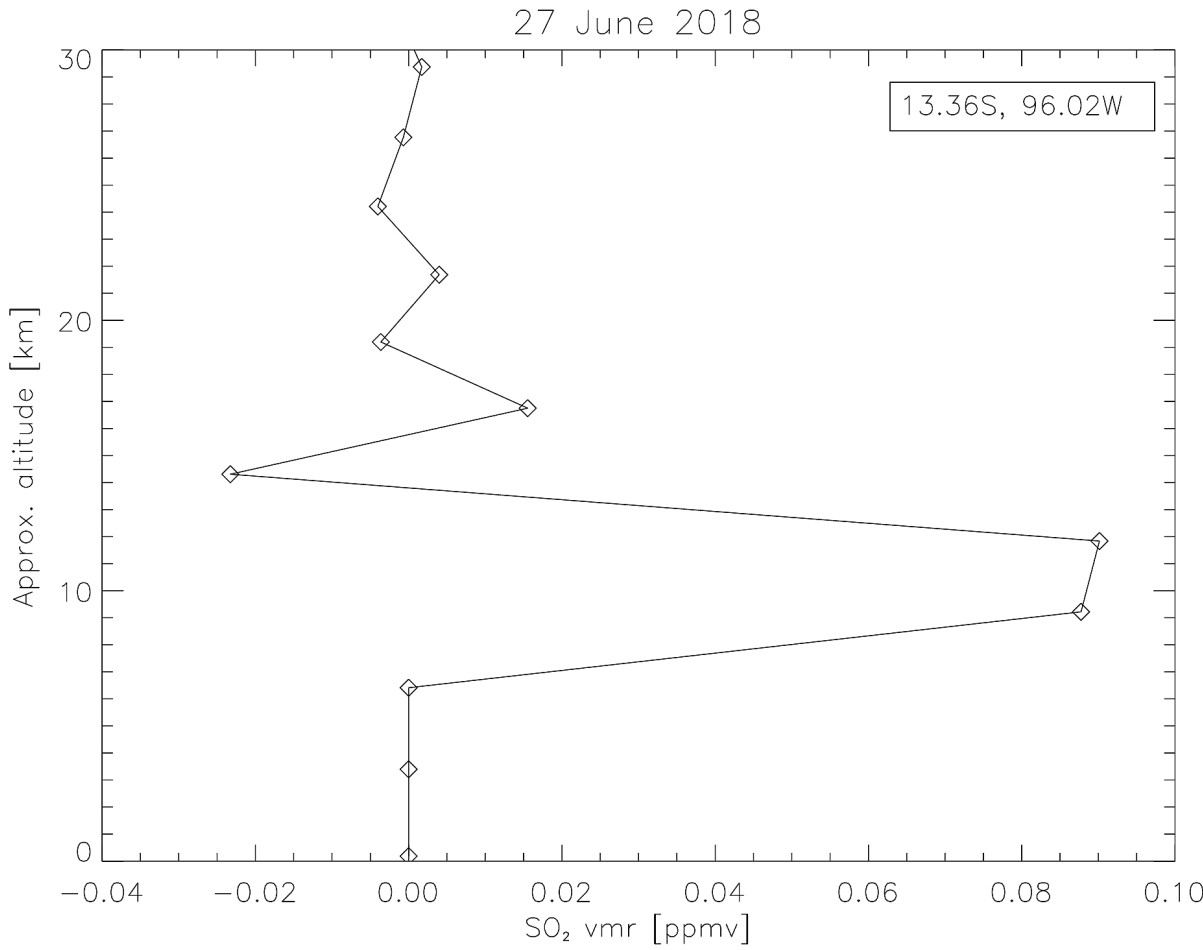

**Figure 11.** MLS SO$_2$ profile measured on 27 February 2018 (at a measurement time of 20:16h UTC) in the extended volcanic plume from the Sierra Negra eruption, clearly showing the presence of an SO$_2$ layer at an altitude range from 9–11 km. The position of the measurements is shown in Fig. 10.

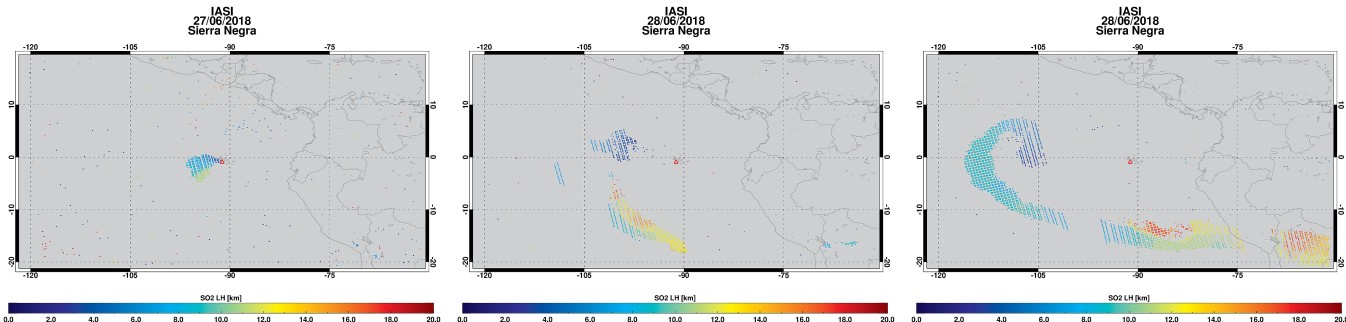

**Figure 12.** MetOp-A and -B IASI SO$_2$ LH results for the Sierra Negra SO$_2$ plume from 27 (left) until 29 June 2018 (right) with overpass times around 02:00h, hence about 6 hours after the TROPOMI measurement.

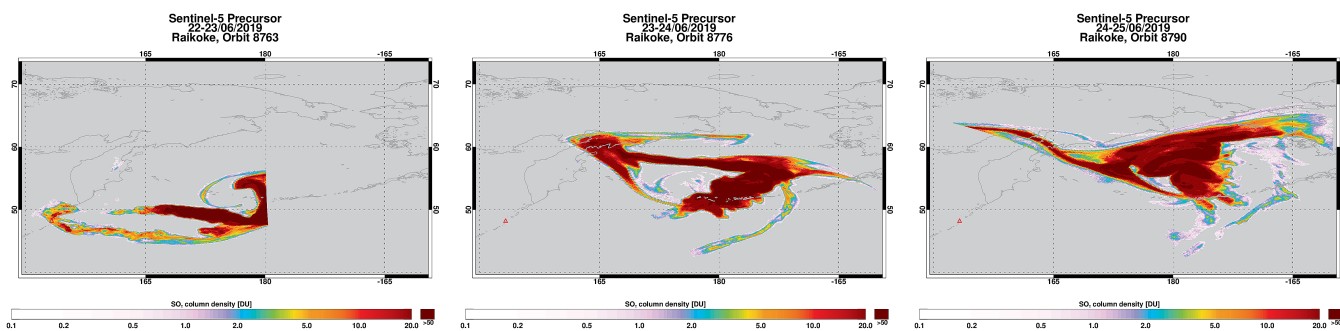

**Figure 13.** SO$_2$ VCD for the TROPOMI measurements of the Raikoke volcanic eruption, measured on 22/23 June 2019 (left), 23/24 June 2019 (center) and 24/25 June 2019 (right), with overpass times around 02:00 UTC (orbit 8763, left), and 00:00h UTC (orbits 8776 and 8790).

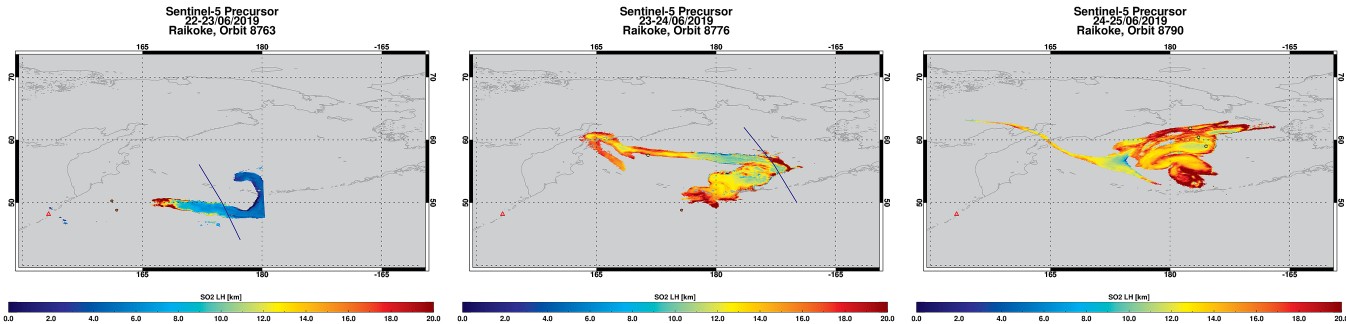

**Figure 14.** SO$_2$ LH retrieved for the TROPOMI orbits shown in Fig. 13. Only pixels with SO$_2$ VCD$\geq$20 DU are shown. Colored circles indicate the MLS/AURA SO$_2$ profile peak height for the results shown in Fig. 15. The blue lines indicate the CALIPSO ground track for the results shown in Fig. 16.

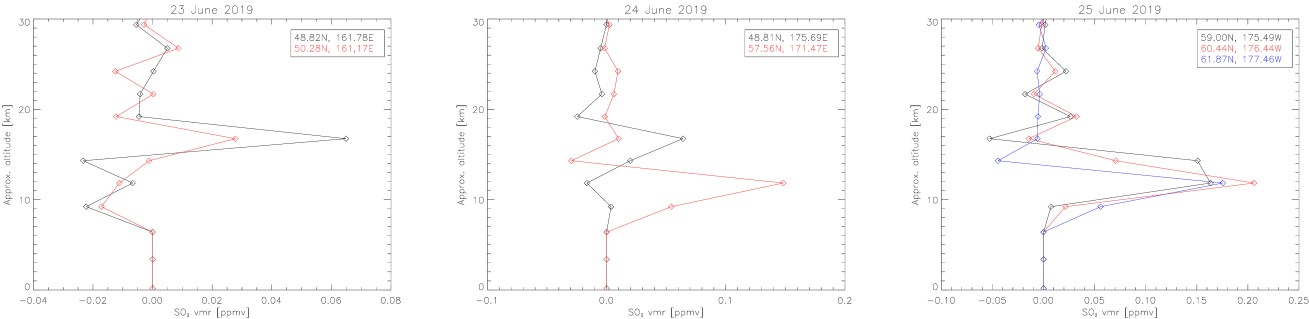

**Figure 15.** MLS SO$_2$ profiles measured on 23 June 2019 (left, overpass around 02:20h UTC), 24 June 2019 (center, overpass around 01:30h UTC), and 25 June 2019 (right, overpass around 00:30h UTC),in the extended volcanic plume from the Raikoke eruption, clearly showing the presence of an SO$_2$ layer at an altitude around 17 km on 23 June, two distinct layers at 12 and 17 km at different positions of the plume on 24 June 2019, and around 12–14 km on 25 June 2019. The position of the measurements is shown in Fig. 14.

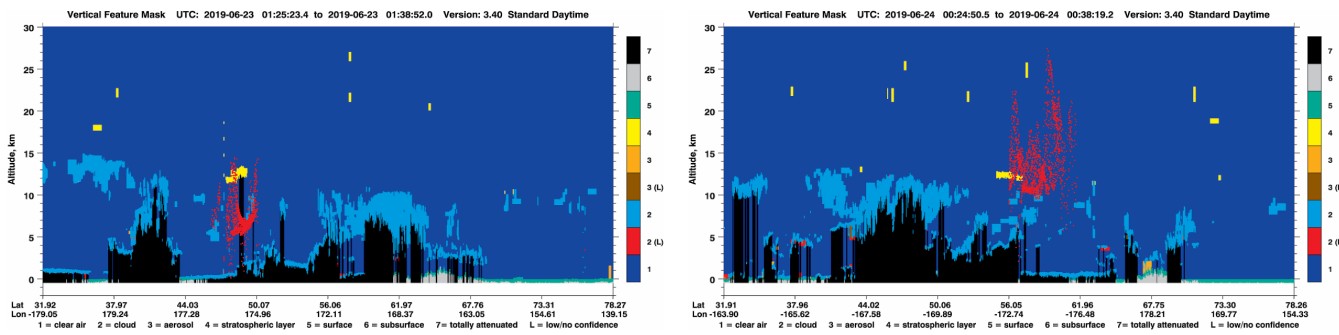

**Figure 16.** CALIOP/CALIPSO vertical feature mask for the measurements of the Raikoke plume on 23 June 2019 around 01:30h UTC (left) and on 24 June 2019 around 00:30h UTC (right). Red dots show the retrieved FP_ILM SO$_2$ LH for pixels with SO$_2$ VCD$\geq$20 DU. The ground track of the CALIPSO measurements is shown in Fig. 14. Note that the classification scheme sometimes cannot pick up the volcanic ash or sulfate aerosol because of competing clouds. CALIPSO image credit: NASA