# Peer review of "SO2 Layer Height retrieval from Sentinel-5 Precursor/TROPOMI using FP\_ILM"

_Atmospheric Measurement Techniques, 2019_

## Referee Comment (RC1) · Anonymous Referee #1 · 14 Mar 2019

This paper introduces a SO2 plume height retrieval algorithm for the high-resolution TROPOMI instrument. The authors outlined the general procedure in developing the algorithm, and also showed example retrievals for some volcanic eruptions observed by TROPOMI. Comparisons were made with IASI retrieved SO2 height, and also CALIPSO measured backscatter. This is an important paper that should be of interest to the broad atmospheric science community. The injection height from volcanic eruptions is a key factor that determines their climate impact. The SO2 plume height can also be useful for aviation safety applications. The paper is generally well written and I would recommend that it be published after the comments below are addressed:

One main concern, for the specific algorithm, is that it uses wavelengths as short as 310 nm from TROPOMI. While these wavelengths provide higher sensitivity to SO2

[Figure]

and SO2 plume height, stray light at these wavelengths also can impact the retrievals. I wonder how the authors address this in their training?

I'm not sure how the 2 km accuracy stated in the abstract is determined. I assume this is based on results in Figures 2-4? On the other hand, the temporal mismatch between TROPOMI and the primary validation dataset (IASI) makes it difficult to support this accuracy estimate.

Also given the difference in TROPOMI and IASI overpass time, some sort of trajectory analysis may help to better link the two retrievals in the comparison.

For NRT applications, the algorithm would require input of O3 VCD? How much time is required to retrieve O3? Also can the authors discuss if the plume height algorithm will run all the time or just be triggered by eruptions?

Page 1, Line 15: oxidation of SO2 also takes place in the troposphere.

Page 3: Line 30: can the authors give some examples of the parameter grids determined by the smart sampling technique?

Page 4, Line 12: Maybe figure 6 should be Figure 1, since it is discussed before all other figures.

Page 6, Line 10: is there a way to determine which rows should be used for training that would provide optimized retrievals? One would assume that pixels with large SO2 VCDs should be used?

Figure 1: it is a bit surprising that the error ca be larger for really high SO2 VCD (close to 1000 DU), do we know why?

Figure 1: it is not very obvious from the figure that high albedo values have a negative impact on the height retrievals – why limit training to albedo < 0.5?

Figures 3 and 4: add shade to mark +/- 2 km from the "real" plume height.

Figure 7: plume height retrievals were also done for pixels with small amount of SO2?

Figure 8: would suggest to only plot TROPOMI retrievals with SO2 > 20 DU.

Figure 9: how much can the ~30 km retrieved plume height be trusted, if the training data only go up to 20 km (Table 1)?

Figure 11: suggest to plot CALIPSO ground track on one of the maps.

[Figure]

---

## Author Comment (AC1) · 4 Apr 2019

First of all, many thanks for the detailed feedback and issues found. We provide feedback to each comment in the following:

One main concern, for the specific algorithm, is that it uses wavelengths as short as 310 nm from TROPOMI. While these wavelengths provide higher sensitivity to SO2 and SO2 plume height, straylight at these wavelengths also can impact the retrievals .I wonder how the authors address this in their training?

- It is correct that straylight can introduce spectral features that may lead to bias in the retrieved SO2 LH, but straylight in Band 3 of TROPOMI (i.e. around 310nm) is very small: According to Kleipool et al (2018), see https://www.atmos-meas-

tech.net/11/6439/2018/amt-11-6439-2018.html table 8 and figure 25, bottom left, the in-band straylight after correction is as low as 0,5% (and thus meets the instrument requirements). Also no evidence for out-of-band-straylight in the UVIS was found by the TROPOMI L1 team (pers. comm. Quintus Kleipool)

I'm not sure how the 2 km accuracy stated in the abstract is determined. I assume this is based on results in Figures 2-4? On the other hand, the temporal mismatch between TROPOMI and the primary validation dataset (IASI) makes it difficult to support this accuracy estimate. Also given the difference in TROPOMI and IASI overpass time, some sort of trajectory analysis may help to better link the two retrievals in the comparison.

- The 2km accuracy is determined by applying the FP_ILM to an independent test dataset with known SO2 LH. Du to missing independent plumeheight data (from ground or other satellites) at exactly the same time as the S5p overpass, no other way to determine the accuracy is possible

- Unfortunately we have only limited experience with trajectory analysis models, but we have performed simple particle trajectory analysis using HYSPLIT in which we have started particles at the IASI LH points and followed them until the S5P overpass time. Based on this 'forecast' we will make a collocation IASI-TROPOMI based on closest distance, and generate statistics (correlation, mean, std, ..) in the updated paper. This however works only for extended plumes, i.e. that of Sierra Negra and Ambae. For Sinabung, we are too close to the (active) source to perform a meaningful colocation.

For NRT applications, the algorithm would require input of O3 VCD? How much time is required to retrieve O3? Also can the authors discuss if the plume height algorithm will run all the time or just be triggered by eruptions?

- Yes, the O3 VCD is required to compensate for the strong spectral interference between SO2 and O3 in the spectral range used for the SO2 LH retrieval. The O3 VCD is an operational TROPOMI product and is also used as input for the operational SO2 VCD retrieval (needed to calculate AMFs). Hence no additional time to retrieve O3

would required in the operational TROPOMI L2 retrieval.

- The SO2 LH algorithm should be triggered by the operational SO2 VCD detection flag, which flags pixel showing an enhanced SO2 signal. This can of course be optimized to trigger the LH retrieval above a certain VCD threshold. We will add a paragraph, discussing the possible implementation in the operational TROPOMI retrieval

Page 1, Line 15: oxidation of SO2 also takes place in the troposphere.

- Indeed. We have corrected the sentence slightly

Page 3: Line 30: can the authors give some examples of the parameter grids determined by the smart sampling technique?

- We will add more information on the parameters grid. However, there was a detailed description in the previous paper from Efremenko et al. 2017

Page 4, Line 12: Maybe figure 6 should be Figure 1, since it is discussed before all other figures.

- You are right. We will show Figure 6 first before the other figures

Page 6, Line 10: is there a way to determine which rows should be used for training that would provide optimized retrievals? One would assume that pixels with large SO2 VCDs should be used?

- In principle, an inversion operator for each single row should be trained, since each row has it's own instrumental characteristic. For the detection of a volcanic SO2 plume, we don't know in advance in which detector row it will be detected, so we made the choice of interpolating the plume height results retrieved for every 50th row to the actual row where it was detected.

Figure 1: it is a bit surprising that the error can be larger for really high SO2 VCD (close to 1000 DU), do we know why?

- Note that the figure shows the results for random parameter choices. The large errors seen for really high SO2 VCD is related to a very high SZA in these cases

Figure 1: it is not very obvious from the figure that high albedo values have a negative impact on the height retrievals – why limit training to albedo < 0.5?

- The large ranges of albedo introduces large variations in the spectra (many refelctions from the surface). This additional variations correlate with the variations due to perturbations in the SO2 VCD and LH. Therefore, to make the algorithm more stable, we restrict the albedo range to the physically relevant cases

Figures 3 and 4: add shade to mark +/- 2 km from the "real" plume height.

- Very good suggestion! We will update the figure accordingly

Figure 7: plume height retrievals were also done for pixels with small amount of SO2?

- Yes, we have applied the FP_ILM to all pixels for which an enhanced SO2 signal was detected (there is a flag in the operational SO2 product) are, hence even for low SO2 VCDs with correspondingly lower SO2 LH accuracies. . .

Figure 8: would suggest to only plot TROPOMI retrievals with SO2 > 20 DU.

- We also thought about this, but then one would only see a few pixels and not the entire extend of the SO2 plume

Figure 9: how much can the âĹij30 km retrieved plume height be trusted, if the training data only go up to 20 km (Table 1)?

- It is correct that the any LH result exceeding the training range has probably a high uncertainty. These cases should be treated with different optimized retrieval operator in an operational environment. In this sense we propose a 2-step retrieval: Once extreme values are detected, we switch to a designated operator.

Figure 11: suggest to plot CALIPSO ground track on one of the maps.

[Figure]

- Good idea, we will overplot the CALIPSO ground track

---

## Referee Comment (RC2) · Anonymous Referee #2 · 15 May 2019

In this paper the authors present an algorithm to retrieve the altitude (or layer height, LH) of volcanic sulfur dioxide (SO2) clouds in near real-time using ultraviolet (UV) satellite data from the Sentinel-5P/TROPOMI instrument. TROPOMI provides the highest spatial resolution UV observations currently available from space. The injection altitude of SO2 during volcanic activity is the main factor determining the climate impact of volcanic eruptions, and can also be used as a reasonable proxy for volcanic ash cloud altitude, which is required for aviation hazard mitigation. Hence accurate retrievals of SO2 altitude are important and of broad interest to the atmospheric and volcano science community. The main advance described in the paper is the adaptation of an existing 'machine learning' SO2 altitude retrieval algorithm (FP_ILM) to the relatively new TROPOMI instrument. The advantage of the FP_ILM algorithm over most exist-

ing SO2 altitude retrievals is the fast processing speed, which allows it to run in near real-time.

Overall I think the paper could be suitable for publication in AMT after some moderate revisions. The structure of the paper could be improved – currently there are many short paragraphs and not all the information is presented in a logical order, and many figures could be improved (see detailed comments below). I do question why the authors only simulated SO2 layer heights up to a maximum of 20 km? Major volcanic eruptions (with the largest potential climate impacts) can inject SO2 to greater altitudes and hence it would be interesting to see how the FP_ILM algorithm would perform in such a scenario, given that the FP_ILM retrievals of SO2 at 20 km altitude appear least accurate for low SO2 VCDs (e.g., Figure 4 suggests that a VCD > 40 DU is needed for accurate retrieval). On a related note, under very high SO2 loadings in a major eruption the ozone (O3) VCD retrievals may be inaccurate (due to SO2 interference), and I assume this would preclude accurate SO2 LH retrieval (since the O3 VCD is a required input). I also find that the stated SO2 LH accuracy of 'better than 2 km for SO2 VCD > 20 DU' is a little exaggerated, especially for higher SO2 LH, e.g., for an SO2 VCD of 20 DU at 20 km, the SO2 LH appears underestimated by ∼5 km in Figure 3.

Another weakness is the validation of the TROPOMI SO2 LH using IASI. Since IASI measurements are not coincident with TROPOMI, only broad conclusions can be drawn from the comparisons. In addition to CALIOP, the authors could explore the use of Microwave Limb Sounder (MLS) SO2 data from the Aura satellite to validate the SO2 LH retrievals.

Specific comments:

P1, L21: there are many different 'flavours' of DOAS algorithm, so writing 'the DOAS algorithm' seems to be a major generalization. Furthermore, there should probably also be a reference to the first multi-spectral Total Ozone Mapping Spectrometer (TOMS)

SO2 retrievals, which used a different approach.

P2, L1: 'fast enough for NRT retrievals' – algorithm speed/computational cost and the timeliness of retrievals are mentioned several times in the paper (e.g., P2, L19-20; P3, L3-5; P3, L17), but there is little quantitative information (I see that there is some information on P6). I would recommend adding a brief discussion to the introduction describing the data latency desired (e.g., for aviation safety and other applications) and estimates of current processing speeds.

P2, L2-4: I think any SO2 algorithm (regardless of whether an AMF is explicitly used) needs to make some assumptions regarding the SO2 vertical distribution (due to the pressure/temperature dependence of SO2 absorption).

P2, L5: accurate AMF calculations could also include parameters such as cloud fraction, surface pressure and surface reflectivity. Also, it is not unique to the 305-335 nm range.

P2, L6: some of paragraph 2 basically restates the previous paragraph; i.e., the SO2 VCD is strongly dependent on the vertical distribution of SO2, as the latter strongly affects the AMF. These paragraphs could be reorganized/combined to clarify the text.

P2, L10: it could be added here (instead of L16) that the usual approach for operational SO2 retrievals (not only from TROPOMI but also other UV sensors) is to assume several different a-priori SO2 vertical distributions and provide VCDs for each. I think it is also important to stress that above ∼5 km or so (i.e., in the upper troposphere and above), the vertical SO2 distribution has relatively little impact on the VCD (although the actual altitude is still of interest of course).

P2, L14: it is not only the number of photons but also the UV wavelengths interacting with the SO2 layer that are influenced by the SO2 layer height.

P2, L16: TROPOMI is first mentioned here, but some key information is provided later on L29 – some reorganization is needed.

P2, L21: Extensive -> Extended.

P2, L22: it is not clear what is meant by 'strong volcanic eruptions'. Eruptions can be relatively weak and still produce high SO2 column amounts, and vice versa.

P2, L26-27: is there a reference to support the statement that IR SO2 height retrievals are more accurate than UV retrievals? I'm not sure that either approach has been extensively validated.

P3, L3: It would be useful to know how often the algorithm needs to be 'trained'. Is re-training necessary if the TROPOMI data quality changes, or for other reasons?

P3, L6: please also cite the first paper on the FP_ILM SO2 LH algorithm here. It might also be useful to briefly summarize the 'improvements' to the algorithm here too.

P3, L24: 'plume profile'.

P4, L10: eight parameters were used to simulate the spectra, but a larger number (10) of PCs is needed to retrieve the layer height. This seems to contradict the authors assertion (P4, L9) that 'fewer parameters' are used to characterize the dataset after the PCA. Some more explanation/clarification may be needed here.

P4, L12: Figure 6 is the first Figure referenced here – in which case the Figures should be reordered.

P4, L16: This paragraph (and also the following one) is probably difficult to follow for anyone not acquainted with neural networks or machine learning. Several new terms are introduced without elaboration (loss function, weight vectors, hidden layers). I recommend that the authors provide more details on the procedure.

P5, L5: shouldn't the O3 VCD also be listed as a direct dependency?

P6, L2: from Figure 4, it appears that the accuracy of SO2 LH retrieval does not significantly improve with increasing SNR for high altitude SO2 LH (20 km). Can the authors explain this?

P6, L23: The IASI data are not the only source of independent SO2 LH data. The Microwave Limb Sounder (MLS) on the NASA/Aura satellite can provide some information on SO2 LH, albeit with limited spatial coverage and vertical resolution. The afternoon MLS overpass is nearly coincident with TROPOMI, and MLS did detect at least some of the eruptions discussed in the paper. I wonder if the authors considered using the MLS data to validate their SO2 LH retrievals?

P6, L26: it would be useful to have at least a few more details on the Ambae volcanic activity (and actually for all the eruptions discussed in the paper), e.g., from the Smithsonian Institution Global Volcanism Program reports.

P6, L30: it is unclear why the plume is 'aged'? See general comment below regarding the SO2 LH map (only pixels with robust LH retrievals should be shown). The two plumes discussed in the text are swamped by areas of blue (low SO2 layer heights which I presume are incorrect due to the generally low SO2 VCDs) and hence hard to see.

P7, L6: need to stress that this is also from Ambae.

P7, L8: acid rain is not usually an issue for stratospheric SO2.

P7, L15: a brief description and reference for the CALIOP instrument is needed (also provide the full name of the sensor).

P7, L16: it should be noted that it is not necessarily the case that SO2 and ash/aerosols are collocated, as gas and ash can separate in volcanic clouds (e.g., as ash falls out to lower altitudes).

Figures:

Figures 3 and 4 are quite similar and could perhaps be combined as one figure.

Figure 5: this figure is a bit cluttered and could perhaps be improved by removing some of the data from the plot, e.g., using just the higher SO2 VCDs.

Figures 7-15: General comment on the SO2 map figures: I would recommend 'zooming in' as much as possible on the SO2 plumes to show the detailed structure (especially Sinabung). Also, I think the SO2 LH plots should only show those TROPOMI pixels with robust SO2 LH retrievals (i.e., SO2 VCD > 10-20 DU or so), since otherwise most of the plots are showing invalid data.

Figure 8: I'm not sure that it is necessary to show both the IASI-A and IASI-B SO2 LH data. Since neither are coincident with TROPOMI, just show the overpass that is closest in time and/or which has the best coverage of the volcanic plume.

Figure 11: the CALIPSO satellite track corresponding to the lidar data in Figure 12 should be shown on the maps.

Figure 12: this figure is also not very clear. I recommend 'zooming in' to the volcanic plume to show the data more clearly, and only plotting the red symbols (SO2 VCD > 20 DU). It is also not clear which features represent the Sinabung volcanic eruption cloud and which are meteorological clouds; this could be highlighted on the plot.

Figure 13: the left-hand panel does not seem to show much if any useful IASI data. Given that there are CALIOP data for this case, perhaps the IASI data are not needed and this figure could be removed.

---

## Author Comment (AC2) · 3 Jun 2019

First of all, many thanks for the detailed feedback and issues found. We provide feedback to each comment in the following:

- In this paper the authors present an algorithm to retrieve the altitude (or layer height,LH) of volcanic sulfur dioxide (SO2) clouds in near real-time using ultraviolet (UV) satellite data from the Sentinel-5P/TROPOMI instrument. TROPOMI provides the highest spatial resolution UV observations currently available from space. The injection altitude of SO2 during volcanic activity is the main factor determining the climate impact of volcanic eruptions, and can also be used as a reasonable proxy for volcanic ash cloud altitude, which is required for aviation hazard mitigation. Hence accurate

retrievals of SO2 altitude are important and of broad interest to the atmospheric and volcano science community. The main advance described in the paper is the adaptation of an existing 'machine learning' SO2 altitude retrieval algorithm (FP_ILM) to the relatively new TROPOMI instrument. The advantage of the FP_ILM algorithm over most existing SO2 altitude retrievals is the fast processing speed, which allows it to run in near real-time.

- Overall I think the paper could be suitable for publication in AMT after some moderate revisions. The structure of the paper could be improved – currently there are many short paragraphs and not all the information is presented in a logical order, and many figures could be improved (see detailed comments below).

–> We will improve both text and figures

- I do question why the authors only simulated SO2 layer heights up to a maximum of 20 km? Major volcanic eruptions (with the largest potential climate impacts) can inject SO2 to greater altitudes and hence it would be interesting to see how the FP_ILM algorithm would perform in such a scenario, given that the FP_ILM retrievals of SO2 at 20 km altitude appear least accurate for low SO2 VCDs (e.g., Figure 4 suggests that a VCD > 40 DU is needed for accurate retrieval).

–> We will extend our training dataset to include SO2 LH up to 30km. We will also show how the NN responds to SO2 LHs not used in the training dataset.

- On a related note, under very high SO2 loadings in a major eruption the ozone (O3) VCD retrievals may be inaccurate (due to SO2 interference), and I assume this would preclude accurate SO2 LH retrieval (since the O3 VCD is a required input).

–> Small errors are expected in O3 VCD and therefore the impact on the SO2 LH retrieval should be minor. See Lerot et al (2013, "Homogenized total ozone data records from the European sensors GOME/ERS-2, SCIAMACHY/Envisat, and GOME-2/MetOp-A"): "The effect is negligible, except for a major volcanic eruption scenario

with SO2 column amounts exceeding 50 DU. In this case, total ozone errors may reach a few percent." -We will show the influence of an inaccurate O3 VCD on the retrieved SO2 LH

- I also find that the stated SO2 LH accuracy of 'better than 2 km for SO2 VCD > 20 DU' is a little exaggerated, especially for higher SO2 LH, e.g., for an SO2 VCD of 20 DU at 20 km, the SO2 LH appears underestimated by âĹij 5 km in Figure 3.

–>We will update the figure, clearly showing the anticipated accuracy of 2km

–>Furthermore we will compare our results to IASI and MLS results by determining the LH at the same overpass time using an dispersion model (i.e. HySplit)

- Another weakness is the validation of the TROPOMI SO2 LH using IASI. Since IASI measurements are not coincident with TROPOMI, only broad conclusions can be drawn from the comparisons. In addition to CALIOP, the authors could explore the use of Microwave Limb Sounder (MLS) SO2 data from the Aura satellite to validate the SO2 LH retrievals.

–>Good point. We will check whether we find MLS measurements for the volcanic cases studied. Unfortunately, IASI is the only other satellite source which we can use for the validation. In order to correct for the overpass time difference between S5P & IASI, we will use a trajectory model (HySplit) to forecast the IASI LH at the S5P overpass time

- P1, L21: there are many different 'flavours' of DOAS algorithm, so writing 'the DOAS algorithm' seems to be a major generalization. Furthermore, there should proba- bly also be a reference to the first multi-spectral Total Ozone Mapping Spectrometer (TOMS) SO2 retrievals, which used a different approach.

–>We will update the text accordingly and add a reference to TOMS and other SO2 retrievals (e.g. OMI PCA)

- P2, L1: 'fast enough for NRT retrievals' – algorithm speed/computational cost and the

timeliness of retrievals are mentioned several times in the paper (e.g., P2, L19-20;P3, L3-5; P3, L17), but there is little quantitative information (I see that there is some information on P6). I would recommend adding a brief discussion to the introduction describing the data latency desired (e.g., for aviation safety and other applications) and estimates of current processing speeds.

–>Very good point. The TROPOMI/S5P NRT data is available 3 hours after sensing. We will add a sentence describing the current processing speed of the operational SO2 VCD retrieval along with an estimation of the extra time needed for the SO2 LH retrieval

- P2, L2-4: I think any SO2 algorithm (regardless of whether an AMF is explicitly used) needs to make some assumptions regarding the SO2 vertical distribution (due to the pressure/ temperature dependence of SO2 absorption).

–>Indeed, we will rephrase the sentence

- P2, L5: accurate AMF calculations could also include parameters such as cloud fraction, surface pressure and surface reflectivity. Also, it is not unique to the 305-335 nm range.

–>Indeed, we will rephrase the sentence

- P2, L6: some of paragraph 2 basically restates the previous paragraph; i.e., the SO2 VCD is strongly dependent on the vertical distribution of SO2, as the latter strongly affects the AMF. These paragraphs could be reorganized/combined to clarify the text.

–>We will reorganize the text

- P2, L10: it could be added here (instead of L16) that the usual approach for operational SO2 retrievals (not only from TROPOMI but also other UV sensors) is to assume several different a-priori SO2 vertical distributions and provide VCDs for each. I think itis also important to stress that above âĹij5 km or so (i.e., in the upper troposphere and above), the vertical SO2 distribution has relatively little impact on the VCD (although the actual altitude is still of interest of course).

–>We will update the text accordingly

- P2, L14: it is not only the number of photons but also the UV wavelengths interacting with the SO2 layer that are influenced by the SO2 layer height.

–>Correct. We will update the text accordingly

- P2, L16: TROPOMI is first mentioned here, but some key information is provided later on L29 – some reorganization is needed.

–>We will reorganize the paper to improve readability

- P2, L21: Extensive -> Extended.

–>We will update the text accordingly

- P2, L22: it is not clear what is meant by 'strong volcanic eruptions'. Eruptions can be relatively weak and still produce high SO2 column amounts, and vice versa.

–>We will use the term Volcanic Explosivity Index (VEI) instead to clarify what is meant here

- P2, L26-27: is there a reference to support the statement that IR SO2 height retrievals are more accurate than UV retrievals? I'm not sure that either approach has been extensively validated.

–>The paper of Clarisse et al. (2014, see https://www.atmos-chem-phys.net/14/3095/2014/) on the Nabro eruption demonstrates that the IR is capable of being very accurate and sensitive to SO2 height when the SO2 columns are low. This paper also presents quite comprehensive validation with CALIPSO. Another (but similar) algorithm by the Carboni et al. (2016, see https://www.atmos-chem-phys.net/16/4343/2016/) shows equally good results. We will add these two references to our text

- P3, L3: It would be useful to know how often the algorithm needs to be 'trained'. Is

re-training necessary if the TROPOMI data quality changes, or for other reasons?

–>The algorithm needs to be re-trained only when large changes in the instrument slit function (ISRF) or SNR occur. Based on the experience from previous UV satellite sensors, large ISRF changes are not expected and, as can be seen from Figure 4, a (moderate) change in the SNR has no big impact on the SO2 LH retrieval results.

- P3, L6: please also cite the first paper on the FP_ILM SO2 LH algorithm here. It might also be useful to briefly summarize the 'improvements' to the algorithm here too.

–>We will add the reference here

- P3, L24: 'plume profile'.

–>We will update the text

- P4, L10: eight parameters were used to simulate the spectra, but a larger number (10) of PCs is needed to retrieve the layer height. This seems to contradict the authors assertion (P4, L9) that 'fewer parameters' are used to characterize the dataset after the PCA. Some more explanation/clarification may be needed here.

–>The 'fewer parameters' refers to the comparison between 10 PCs and the corresponding 161 spectral points of the SO2 LH fitting window.

- P4, L12: Figure 6 is the first Figure referenced here – in which case the Figures should be reordered.

–>We will reorder the Figures accordingly.

- P4, L16: This paragraph (and also the following one) is probably difficult to follow for anyone not acquainted with neural networks or machine learning. Several new terms are introduced without elaboration (loss function, weight vectors, hidden layers).

–>We will update the paragraph and add more information about NNs and machine learning

[Figure]

- I recommend that the authors provide more details on the procedure.

–>We will update the text accordingly

- P5, L5: shouldn't the O3 VCD also be listed as a direct dependency?

–>That's correct, we will add it to the list

- P6, L2: from Figure 4, it appears that the accuracy of SO2 LH retrieval does not significantly improve with increasing SNR for high altitude SO2 LH (20 km). Can the authors explain this?

–>The SO2 LH retrieval is more sensitive to higher-altitude plumes than to low-altitude plumes. Therefore, increasing SNR will improve the accuracy for low plumes but not so much for higher plumes.

- P6, L23: The IASI data are not the only source of independent SO2 LH data. The Microwave Limb Sounder (MLS) on the NASA/Aura satellite can provide some information on SO2 LH, albeit with limited spatial coverage and vertical resolution. The afternoon MLS overpass is nearly coincident with TROPOMI, and MLS did detect at least some of the eruptions discussed in the paper. I wonder if the authors considered using the MLS data to validate their SO2 LH retrievals?

–>Good point. We will also include MLS results for Sinabung

- P6, L26: it would be useful to have at least a few more details on the Ambae volcanic activity (and actually for all the eruptions discussed in the paper), e.g., from the Smithsonian Institution Global Volcanism Program reports.

–>Ok, we will add more information on the volcanoes presented in this paper

- P6, L30: it is unclear why the plume is 'aged'? See general comment below regarding the SO2 LH map (only pixels with robust LH retrievals should be shown). The two plumes discussed in the text are swamped by areas of blue (low SO2 layer heights which I presume are incorrect due to the generally low SO2 VCDs) and hence hard to

see.

–>With 'aged' we meant that the plume has already travelled from its source after the eruption until the S5P measurement took place. Actually we discovered that the image shows the plume with a time difference of 24 hours due to the date line: The plume in Fig.9 close to the volcano (around 180E) is the plume observed in the morning of 27 June, whereas the part of the plume around 150W is the same SO2 cloud but observed on the evening of 27 June. We will hence update the figure accordingly. In the revised paper we will only show results for SO2 close to 20DU.

- P7, L6: need to stress that this is also from Ambae.

–>This sentence belongs to the Ambae section, so this should be clear. But we will make this more clear in the text

- P7, L8: acid rain is not usually an issue for stratospheric SO2.

–>Indeed. We will correct the sentence

- P7, L15: a brief description and reference for the CALIOP instrument is needed (also provide the full name of the sensor).

–>We will add a brief description for CALIOP

- P7, L16: it should be noted that it is not necessarily the case that SO2 and ash/aerosols are collocated, as gas and ash can separate in volcanic clouds (e.g., as ash falls out to lower altitudes).

–>Correct. We will point this out in the revised paper

Figures:

- Figures 3 and 4 are quite similar and could perhaps be combined as one figure.

–>That's correct. We will combine the figures

- Figure 5: this figure is a bit cluttered and could perhaps be improved by removing

some of the data from the plot, e.g., using just the higher SO2 VCDs.

–>We will improve the figure

- Figures 7-15: General comment on the SO2 map figures: I would recommend 'zooming in' as much as possible on the SO2 plumes to show the detailed structure (especially Sinabung). Also, I think the SO2 LH plots should only show those TROPOMI pixels with robust SO2 LH retrievals (i.e., SO2 VCD > 10-20 DU or so), since otherwise most of the plots are showing invalid data.

–>We will make zoom-ins. The original intend was to show the same Lat-Lon range as when overplotting the IASI data.

- Figure 8: I'm not sure that it is necessary to show both the IASI-A and IASI-B SO2 LH data. Since neither are coincident with TROPOMI, just show the overpass that is closest in time and/or which has the best coverage of the volcanic plume.

–>Good point. Another idea would be to use the IASI data as input to a transport model (e.g. HySplit) and forecast the plume movement for the S5P overpass time to perform the comparison

- Figure 11: the CALIPSO satellite track corresponding to the lidar data in Figure 12 should be shown on the maps.

–>We will add the ground track to the figure

- Figure 12: this figure is also not very clear. I recommend 'zooming in' to the volcanic plume to show the data more clearly, and only plotting the red symbols (SO2 VCD > 20 DU). It is also not clear which features represent the Sinabung volcanic eruption cloud and which are meteorological clouds; this could be highlighted on the plot.

–>We will update the plot and zoom-in to the volcanic plume

- Figure 13: the left-hand panel does not seem to show much if any useful IASI data. Given that there are CALIOP data for this case, perhaps the IASI data are not needed

[Figure]

and this figure could be removed.

–>Good point, we will remove the left panel

---

## Author Response (AR1)

**Detailed point-by-point response to all referee comments**

In the revised version of the manuscript attached below we have highlighted changes in red color, changes related to the comments from reviewer #1 in green and changes related to comments from reviewer #2 in blue. Hereafter we first provide an overview about the general changes made to the manuscript before we address the comments from the referees.

**General changes**

All general changes are highlighted in red in the revised manuscript

In general, based on the referee comments, we have improved and slightly restructured the text.

In agreement with the editor, we have removed the entire Ambae section since IASI was the only reference products to compare our SO2 LH results with and the SO2 plume was pretty weak – only a few pixels exceeded our threshold criterion of SO2 VCD > 20DU. We also did not find any CALIPSO or MLS data that could be used for the verification. First of all because there was no ash present and the plume was probably too weak and too low to be detected by MLS. Unfortunately the difference in the overpass times of TROPOMI and IASI were too big as to provide a meaningful comparison. Even with a trajectory model (HySPLIT) we were not able to correct for the different overpass times. This might be related to our limited knowledge of using trajectory models. We have therefore decided to expand our section on Sinabung and Sierra Negra and in addition show results of the very recent Raikoke eruption where we have several MLS, CALIPSO and IASI datasets available for validation.

We have added a new section that describes how the FP_ILM algorithm could be integrated into the operational TROPOMI SO2 retrieval

**Changes made based on review of anonymous referee #1 ()**

All changes related to this review are highlighted in green in the revised manuscript. The response to the referee is structured in the following sequence: (1) comment from referee, (2) author's response, (3) author's change in the manuscript

1.1 One main concern, for the specific algorithm, is that it uses wavelengths as short as 310 nm from TROPOMI. While these wavelengths provide higher sensitivity to SO2 and SO2 plume height, straylight at these wavelengths also can impact the retrievals .I wonder how the authors address this in their training?

1.2 It is correct that straylight can introduce spectral features that may lead to bias in the retrieved SO2 LH, but straylight in Band 3 of TROPOMI (i.e. around 310nm) is very small:
According to Kleipool et al (2018), see https://www.atmos-meas-tech.net/11/6439/2018/amt-11-6439-2018.html table 8 and figure 25, bottom left, the in-band straylight after correction is as low as 0,5% (and thus meets the instrument requirements). Also no evidence for out-of-band-straylight in the UVIS was found by the TROPOMI L1 team (pers. comm. Quintus Kleipool)

1.3 We have added a comment on instrumental straylight in Sect. 3 ("Dependencies")

2.1 I'm not sure how the 2 km accuracy stated in the abstract is determined. I assume this is based on results in Figures 2-4? On the other hand, the temporal mismatch between TROPOMI and the primary validation dataset (IASI) makes it difficult to support this accuracy estimate. Also given the difference in TROPOMI and IASI overpass time, some sort of trajectory analysis may help to better link the two retrievals in the comparison.

2.2 The 2km accuracy is determined by applying the FP_ILM to an independent test dataset with known SO2 LH. Du to missing independent plumeheight data (from ground or other satellites) at exactly the same time as the S5p overpass, no other way to determine the accuracy is possible Unfortunately we have only limited experience with trajectory analysis models, but we have performed simple particle trajectory analysis using HYSPLIT in which we have started particles at the IASI LH points and followed them until the S5P overpass time. Based on this 'forecast' we will make a collocation IASI-TROPOMI based on closest distance, and generate statistics (correlation, mean, std, ..) in the updated paper. This however works only for extended plumes, i.e. that of Sierra Negra and Ambae. For Sinabung, we are too close to the (active) source to perform a meaningful colocation.

2.3 We have added a comment on how we come to the occlusion of achieving the 2km at all occurences in the text.

3.1 For NRT applications, the algorithm would require input of O3 VCD? How much time is required to retrieve O3? Also can the authors discuss if the plume height algorithm will run all the time or just be triggered by eruptions?

3.2 Yes, the O3 VCD is required to compensate for the strong spectral interference between SO2 and O3 in the spectral range used for the SO2 LH retrieval. The O3 VCD is an operational TROPOMI product and is also used as input for the operational SO2 VCD retrieval (needed to calculate AMFs). Hence no additional time to retrieve O3 would be required in the operational TROPOMI L2 retrieval.
The SO2 LH algorithm should be triggered by the operational SO2 VCD detection flag, which flags pixel showing an enhanced SO2 signal. This can of course be optimized to trigger the LH retrieval above a certain VCD threshold. We will add a paragraph, discussing the possible implementation in the operational TROPOMI retrieval

3.3 We have added a new section in which we discuss a possible implementation into the operational S5P environment, see Section 5 "Implementation in an operational environment"

4.1 Page 1, Line 15: oxidation of SO2 also takes place in the troposphere.

4.2 Indeed. We have corrected the sentence slightly

4.3 We have corrected the sentence in P1 L16

5.1 Page 3: Line 30: can the authors give some examples of the parameter grids determined by the smart sampling technique?

5.2 Copy paragraph from previous paper (?)

5.3 We have added a reference to the previous paper on P4 L8, where this is explained in detail in order to avoid repeating the same examples. Also a reference to the original paper presenting this technique has been added

6.1 Page 4, Line 12: Maybe figure 6 should be Figure 1, since it is discussed before all other figures.

6.2 You are right. We will show Figure 6 first before the other figures

6.3 Figure 6 has been moved to the front

7.1 Page 6, Line 10: is there a way to determine which rows should be used for training that would provide optimized retrievals? One would assume that pixels with large SO2 VCDs should be used?

7.2 In principle, an inversion operator for each single row should be trained, since each row has it's own instrumental characteristic. For the detection of a volcanic SO2 plume, we don't know in advance in which detector row it will be detected, so we made the choice of interpolating the plume height results retrieved for every 50$^{th}$ row to the actual row where it was detected.

7.3 We have added a comment to P5 L19

8.1 Figure 1: it is a bit surprising that the error can be larger for really high SO2 VCD (close to 1000 DU), do we know why?

8.2 Note that the figure shows the results for random parameter choices. The large errors seen for really high SO2 VCD is related to a very high SZA in these cases

8.3 The figure has been updated (it is now Figure 2), now showing the dependencies for the first an optimized training test data

9.1 Figure 1: it is not very obvious from the figure that high albedo values have a negative impact on the height retrievals – why limit training to albedo < 0.5?

9.2 The large ranges of albedo introduces large variations in the spectra (many refelctions from the surface). This additional variations correlate with the variations due to perturbations in the SO2 VCD and LH. Therefore, to make the algorithm more stable, we restrict the albedo range to the physically relevant cases.

9.3 We added a sentence to clarify our approach in Section 3 ("Dependencies")

10.1 Figures 3 and 4: add shade to mark +/- 2 km from the "real" plume height.

10.2 Very good suggestion!

10.3 We have updated the Figures, now showing the suggested shade to mark the 2km uncertainty

11.1 Figure 7: plume height retrievals were also done for pixels with small amount of SO2?

11.2 Yes, we have applied the FP_ILM to all pixels for which an enhanced SO$_2$ signal was detected (there is a flag in the operational SO2 product) are, hence even for low SO2 VCDs with correspondingly lower SO2 LH accuracies

11.3 We have updated all figures such that only pixels with VCD>20DU are shown in the LH plots

12.1 Figure 8: would suggest to only plot TROPOMI retrievals with SO2 > 20 DU.

12.2 We also thought about this, but then one would only see a few pixels and not the entire extend of the SO2 plume

12.3 We have removed the entire Ambae section. Nevertheless, for the final version we have decided to plot in all cases the entire plume for the VCD plot, but only the SO2 LH for pixels with >20DU

13.1    Figure 9: how much can the ~30 km retrieved plume height be trusted, if the training data only go up to 20 km (Table 1)?

13.2    It is correct that the any LH result exceeding the training range has probably a high uncertainty. These cases should be treated with different optimized retrieval operator in an operational environment. In this sense we propose a 2-step retrieval: Once extreme values are detected, we switch to a designated operator.

13.3    We have removed the entire Ambae section, since we did not find a good match between Tropomi and IASI (too long time difference), which we were not able to correct for using HySplit updated the figures accordingly. Also other datasets (CALIPSO, MLS) did not provide any measurements for the Ambae eruptions. Instead, we added (in agreement with the editor) a new subsection on the Raikoke volcanic eruption, for which we have an extended set of independent sources of the LH

14.1    Figure 11: suggest to plot CALIPSO ground track on one of the maps.

14.2    Good idea, we will overplot the CALIPSO ground track

14.3    In all SO2 LH plot we are now plotting the CALIPSO ground tracks, if applicable

All changes related to this review are highlighted in blue in the revised manuscript. The response to the referee is structured in the following sequence: (1) comment from referee, (2) author's response, (3) author's change in the manuscript

1.1 Overall I think the paper could be suitable for publication in AMT after some moderate revisions. The structure of the paper could be improved – currently there are many short paragraphs and not all the information is presented in a logical order, and many figures could be improved (see detailed comments below).

1.2 We will improve both text and figures

1.3 The entire text has been improved and partly restructured (see blue highlighted text) and the figures now show zoom-ins.

2.1 I do question why the authors only simulated SO2 layer heights up to a maximum of 20 km? Major volcanic eruptions (with the largest potential climate impacts) can inject SO2 to greater altitudes and hence it would be interesting to see how the FP_ILM algorithm would perform in such a scenario, given that the FP_ILM retrievals of SO2 at 20 km altitude appear least accurate for low SO2 VCDs (e.g., Figure 4 suggests that a VCD > 40 DU is needed for accurate retrieval).

2.2 We will extend our training dataset to include SO2 LH up to 30km. We will also show how the NN responds to SO2 LHs not used in the training dataset.

2.3 The training dataset has been extended to cover also SO2LH up to 30km. In order to optimize our algorithm, we however have restricted our training to use only LHs up to 25km (see Table 1). Although also higher LHs from strong volcanic eruptions occur sometimes, using a broad training data range also has an influence on the accuracy of the retrieval, since the algorithm has to cope with this broad range. To a limited extend however, the FP_ILM is also able to extrapolate to an untrained parameter range, however with significantly decreasing accuracy. We have added a comment in Sect. 3 ("Dependencies")

3.1 On a related note, under very high SO2 loadings in a major eruption the ozone (O3) VCD retrievals may be inaccurate (due to SO2 interference), and I assume this would preclude accurate SO2 LH retrieval (since the O3 VCD is a required input).

3.2 Small errors are expected in O3 VCD and therefore the impact on the SO2 LH retrieval should be minor. See Lerot et al (2013, "Homogenized total ozone data records from the European sensors GOME/ERS-2, SCIAMACHY/Envisat, and GOME-2/MetOp-A"): "The effect is negligible, except for a major volcanic eruption scenario with SO2 column amounts exceeding 50 DU. In this case, total ozone errors may reach a few percent."

3.3 We have added a comment on the dependency of our SO2LH retrieval on error in the O3VCD in Sect. 3 ("Dependencies")

4.1 I also find that the stated SO2 LH accuracy of 'better than 2 km for SO2 VCD > 20 DU' is a little exaggerated, especially for higher SO2 LH, e.g., for an SO2 VCD of 20 DU at 20 km, the SO2 LH appears underestimated by ~ 5 km in Figure 3.

4.2 We will update the figure, clearly showing the anticipated accuracy of 2km. Furthermore we will compare our results to IASI and MLS results by determining the LH at the same overpass time using an dispersion model (i.e. HySplit)

4.3 We have updated Fig. 3, now showing much more test data points for pre-defined LHs. This figure also proves that we reach the 2km uncertainty using this independent test dataset. We have decided to not use HySPLIT to compare TROPOMI and IASI data since our experience with trajectory models in very limited. We have therefore performed comparisons with MLS and CALIPSO data and only compare to IASI data on a qualitative level.

5.1 Another weakness is the validation of the TROPOMI SO2 LH using IASI. Since IASI measurements are not coincident with TROPOMI, only broad conclusions can be drawn from the comparisons. In addition to CALIOP, the authors could explore the use of Microwave Limb Sounder (MLS) SO2 data from the Aura satellite to validate the SO2 LH retrievals.

5.2 Good point. We will check whether we find MLS measurements for the volcanic cases studied. Unfortunately, IASI is the only other satellite source which we can use for the validation. In order to correct for the overpass time difference between S5P & IASI, we will use a trajectory model (HySplit) to forecast the IASI LH at the S5P overpass time

5.3 We have added comparisons with MLS data and found additional CALIPSO overpasses that we could use for comparison. We decided to not use HySPLIT to correct for the overpass time difference between TROPOMI and IASI due to our limited experience with trajectory models. Comparisons with IASI data is used for a qualitative validation on the accuracy of retrieved SO2 LHs from TROPOMI measurement

6.1. P1, L21: there are many different 'flavours' of DOAS algorithm, so writing 'the DOAS algorithm' seems to be a major generalization. Furthermore, there should probably also be a reference to the first multi-spectral Total Ozone Mapping Spectrometer (TOMS) SO2 retrievals, which used a different approach.

6.2. We will update the text accordingly and add a reference to TOMS and other SO2 retrievals (e.g. OMI PCA)

6.3. We have updated the text (see P2, L1) and now also list the PCA and Krueger-Kerr algorithms to retrieve SO2

7.1 P2, L1: 'fast enough for NRT retrievals' – algorithm speed/computational cost and the timeliness of retrievals are mentioned several times in the paper (e.g., P2, L19-20;P3, L3-5; P3, L17), but there is little quantitative information (I see that there is some information on P6). I would recommend adding a brief discussion to the introduction describing the data latency desired (e.g., for aviation safety and other applications) and estimates of current processing speeds.

7.2 Very good point. The TROPOMI/S5P NRT data is available 3 hours after sensing. We will add a sentence describing the current processing speed of the operational SO2 VCD retrieval along with an estimation of the extra time needed for the SO2 LH retrieval

7.3 We have added the new Section 5 "Implementation in an operational environment" that describes how the FP_ILM could be intergated in the operational TROPOMI SO2 retrieval and also provide detailed information about the speed of the algorithm with respect to current processing speeds.

8.1 P2, L2-4: I think any SO2 algorithm (regardless of whether an AMF is explicitly used) needs to make some assumptions regarding the SO2 vertical distribution (due to the pressure/ temperature dependence of SO2 absorption).

8.2 Indeed, we will rephrase the sentence

8.3 We have rephrased the sentence, see P2 L3-5

9.1 P2, L5: accurate AMF calculations could also include parameters such as cloud fraction, surface pressure and surface reflectivity. Also, it is not unique to the 305-335 nm range.

9.2 Indeed, we will rephrase the sentence

9.3 We have rephrased the sentence, see P2 L6

10.1 P2, L6: some of paragraph 2 basically restates the previous paragraph; i.e., the SO2 VCD is strongly dependent on the vertical distribution of SO2, as the latter strongly affects the AMF. These paragraphs could be reorganized/combined to clarify the text.

10.2 We will reorganize the text

10.3 We have restructured the text for clarification

11.1 P2, L10: it could be added here (instead of L16) that the usual approach for operational SO2 retrievals (not only from TROPOMI but also other UV sensors) is to assume several different a-priori SO2 vertical distributions and provide VCDs for each. I think itis also important to stress that above ~5 km or so (i.e., in the upper troposphere and above), the vertical SO2 distribution has relatively little impact on the VCD (although the actual altitude is still of interest of course).

11.2 We will update the text accordingly

11.3 The text has been updated, see P2, L15-18

12.1 P2, L14: it is not only the number of photons but also the UV wavelengths interacting with the SO2 layer that are influenced by the SO2 layer height.

12.2 Correct. We will update the text accordingly

12.3 We have updated the text accordingly

13.1 P2, L16: TROPOMI is first mentioned here, but some key information is provided later on L29 – some reorganization is needed.

13.2 We will reorganize the paper to improve readability

13.3 We have removed the TROPOMI reference here (see comment 11 above)

14.1 P2, L21: Extensive -> Extended.

14.2 We will update the text accordingly

14.3 The text has been updated

15.1 P2, L22: it is not clear what is meant by 'strong volcanic eruptions'. Eruptions can be relatively weak and still produce high SO2 column amounts, and vice versa.

15.2 We will use the term Volcanic Explosivity Index (VEI) instead to clarify what is meant here

15.3    We are now using the term VEI throughout the paper to characterize the strength of the volcanic eruptions

16.1    P2, L26-27: is there a reference to support the statement that IR SO2 height retrievals are more accurate than UV retrievals?   I'm not sure that either approach has been extensively validated.

16.2    The paper of Clarisse et al. (2014, see https://www.atmos-chem-phys.net/14/3095/2014/) on the Nabro eruption demonstrates that the IR is capable of being very accurate and sensitive to SO2 height when the SO2 columns are low. This paper also presents quite comprehensive validation with CALIPSO. Another (but similar) algorithm by the Carboni et al. (2016, see https://www.atmos-chem-phys.net/16/4343/2016/) shows equally good results. We will add these two references to our text

16.3    Both references have been added to the text (see P2, L30)

17.1    P3, L3:  It would be useful to know how often the algorithm needs to be 'trained'.  Is re-training necessary if the TROPOMI data quality changes, or for other reasons?

17.2    The algorithm needs to be re-trained only when large changes in the instrument slit function (ISRF) or SNR occur. Based on the experience from previous UV satellite sensors, large ISRF changes are not expected and, as can be seen from Figure 4, a (moderate) change in the SNR has no big impact on the SO2 LH retrieval results.

17.3    We have added a comment on P5,L21

18.1    P3, L6: please also cite the first paper on the FP_ILM SO2 LH algorithm here. It might also be useful to briefly summarize the 'improvements' to the algorithm here too.

18.2    We will add the reference here

18.3    We have added the reference as well as a description about the improvements on P3, L22-26

19.1    P3, L24: 'plume profile'.

19.2    We will update the text

19.3    We have updated the text

20.1    P4, L10: eight parameters were used to simulate the spectra, but a larger number (10) of PCs is needed to retrieve the layer height.  This seems to contradict the authors assertion (P4, L9) that 'fewer parameters' are used to characterize the dataset after the PCA. Some more explanation/clarification may be needed here.

20.2    The 'fewer parameters' refers to the comparison between 10 PCs and the corresponding 161 spectral points of the SO2 LH fitting window.

20.3    We have clarified our approach in Sect. 2

21.1    P4, L12: Figure 6 is the first Figure referenced here – in which case the Figures should be reordered.

21.2    We will reorder the Figures accordingly.

21.3    Figure 6 has now moved to Figure 1

22.1     P4, L16: This paragraph (and also the following one) is probably difficult to follow for anyone not acquainted with neural networks or machine learning. Several new terms are introduced without elaboration (loss function, weight vectors, hidden layers). I recommend that the authors provide more details on the procedure.

22.2     We will update the paragraph and add more information about NNs and machine learning

22.3     We have updated the text and improve more detailed information about the NN approach in Sect. 2

23.1     P5, L5: shouldn't the O3 VCD also be listed as a direct dependency?

23.2     That's correct, we will add it to the list

23.3     O3 VCD has been added to the list

24.1     P6, L2: from Figure 4, it appears that the accuracy of SO2 LH retrieval does not significantly improve with increasing SNR for high altitude SO2 LH (20 km). Can the authors explain this?

24.2     The SO2 LH retrieval is more sensitive to higher-altitude plumes than to low-altitude plumes. Therefore, increasing SNR will improve the accuracy for low plumes but not so much for higher plumes.

24.3     The SNR has only a minor impact on our SO2LH accuracy, since we effectively filter out noise using the PCA approach and using only the first 10 PC scores. We only see an influence on the results for very low SNR < 500. We have added a comment in Sect. 3, P6 24-27.

25.1     P6, L23:  The IASI data are not the only source of independent SO2 LH data.  The Microwave Limb Sounder (MLS) on the NASA/Aura satellite can provide some information on SO2 LH, albeit with limited spatial coverage and vertical resolution.   The afternoon MLS overpass is nearly coincident with TROPOMI, and MLS did detect at least some of the eruptions discussed in the paper. I wonder if the authors considered using the MLS data to validate their SO2 LH retrievals?

25.2     Good point. We will also include  MLS results for Sinabung

25.3     We have added detailed MLS results for Sinabung, Sierra Negra and Raikoke

26.1     P6,  L26:  it would be useful to have at least a few more details on the Ambae volcanic activity (and actually for all the eruptions discussed in the paper), e.g., from the Smithsonian Institution Global Volcanism Program reports.

26.2     Ok, we will add more information on the volcanoes presented in this paper

26.3     As already described in the general comments above, we have removed the entire Ambae section. However, for all other volcanoes, we have added more information on the volcano itself

27.1     P6, L30: it is unclear why the plume is 'aged'? See general comment below regarding the SO2 LH map (only pixels with robust LH retrievals should be shown).   The two plumes discussed in the text are swamped by areas of blue (low SO2 layer heights which I presume are incorrect due to the generally low SO2 VCDs) and hence hard to see.

27.2     With 'aged' we mean that the plume has already travelled from its source after the eruption until the S5P measurement took place. Actually we discovered that the image shows the plume with a time difference of 24 hours due to the date line: The plume in Fig.9 close to the volcano (around 180E) is the plume observed in the morning of 27 June, whereas the part of the plume around 150W

is the same SO2 cloud but observed on the evening of 27 June. We will hence update the figure accordingly. In the revised paper we will only show results for SO2 close to 20DU.

27.3    The Ambae section has been removed. We now only show SO2LH maps for SO2>20DU.

28.1    P7, L6: need to stress that this is also from Ambae.

28.2    This sentence belongs to the Ambae section, so this should be clear. But we will make this more clear in the text

28.3    The Ambae section has been removed

29.1    P7, L8: acid rain is not usually an issue for stratospheric SO2.

29.2    Indeed. We will correct the sentence

29.3    The Ambae section has been removed

30.1    P7, L15: a brief description and reference for the CALIOP instrument is needed (also provide the full name of the sensor).

30.2    We will add a brief description for CALIOP

30.3    Details on CALIOP have been added to Sect. 4 (see P7, L26)

31.1    P7, L16: it should be noted that it is not necessarily the case that SO2 and ash/aerosols are collocated, as gas and ash can separate in volcanic clouds (e.g., as ash falls out to lower altitudes).

31.2    Correct. We will point this out in the revised paper

31.3    We have added a comment on P7 L27

32.1    Figures 3 and 4 are quite similar and could perhaps be combined as one figure.

32.2    That's correct. We will combine the figures

32.3    We have updated Figure 3 & 4 showing more datapoints from the test dataset. A combination of the figures would result in too many datapoints

33.1    Figure 5: this figure is a bit cluttered and could perhaps be improved by removing some of the data from the plot, e.g., using just the higher SO2 VCDs.

33.2    We will improve the figure

33.3    We have improved the figure by using smaller symbols

34.1    Figures 7-15: General comment on the SO2 map figures: I would recommend 'zooming in' as much as possible on the SO2 plumes to show the detailed structure (especially Sinabung). Also, I think the SO2 LH plots should only show those TROPOMI pixels with robust SO2 LH retrievals (i.e., SO2 VCD > 10-20 DU or so), since otherwise most of the plots are showing invalid data.

34.2    We will make zoom-ins. The original intend was to show the same Lat-Lon range as when overplotting the IASI data.

34.3    We have updated all SO2 maps and zoom-in to the volcanic plume(s)

35.1    Figure 8:  I'm not sure that it is necessary to show both the IASI-A and IASI-B SO2 LH data.  Since neither are coincident with TROPOMI, just show the overpass that is closest in time and/or which has the best coverage of the volcanic plume.

35.2    Good point. Another idea would be to use the IASI data as input to a transport model (e.g. HySplit) and forecast the plume movement for the S5P overpass time to perform the comparison

35.3    We have removed the Ambae section

36.1    Figure 11:  the CALIPSO satellite track corresponding to the lidar data in Figure 12 should be shown on the maps.

36.2    We will add the ground track to the figure

36.3    We have added CALIPSO ground tracks to all SO2LH figures when we compare with the CALIPSO data

37.1    Figure 12:  this figure is also not very clear.  I recommend 'zooming in' to the volcanic plume to show the data more clearly, and only plotting the red symbols (SO2 VCD > 20 DU). It is also not clear which features represent the Sinabung volcanic eruption cloud and which are meteorological clouds; this could be highlighted on the plot.

37.2    We will update the plot and zoom-in to the volcanic plume

37.3    We are now only showing SO2 LH for SO2 > 20DU. As we are not that familiar with CALIPSO L1 data plotting we prefer to stick to the official CALIPSO images from NASA and simply overplot our data without zooming in. We have also decided to show the Vertical Feature Mask instead of the attenuation signal, making it easier to identify different features

38.1    Figure 13:  the left-hand panel does not seem to show much if any useful IASI data. Given that there are CALIOP data for this case, perhaps the IASI data are not needed and this figure could be removed.

38.2    Good point, we will remove the left panel

38.3    We have decided to not show IASI data for the Sinabung eruption

[revised manuscript text omitted]

---

## Author Response (AR2)

**Detailed point-by-point response to the referee comment**

First of all, we would like to thank all reviewers for nicely characterizing our work, its relevance, and for helping to improve the readability of the paper.

We have followed the recommendation of the reviewer to change p.7 line 6 (end of section 3) from 'and hence the error on the SO2 LH is negligible' to 'which means that the error on the SO2 LH is still negligible'.

We have furthermore updated the Acknowledgements section, to acknowledge support from DLR and BIRA for the S5P L2 files. We are now also acknowledging support from the EUNADICS-AV project for the development of the $SO_2$ LH algorithm. Finally we have added some text about TROPOMI and S5P here.

In the revised version of the manuscript attached below we have highlighted changes in red color

[revised manuscript text omitted]